# CRRL: Learning Channel-invariant Neural Representations for High-performance Cross-day Decoding

**Xianhan Tan**[1,2,3 †]**, Binli Luo**[4 †]**, Yu Qi**[1,2,3,5]*,**Yueming Wang**[1]
[1]MOE Frontier Science Center for Brain Science and Brain-machine Integration, Zhejiang University
[2]Affiliated Mental Health Center & Hangzhou Seventh People's Hospital, Zhejiang University
[3]College of Computer Science and Technology, Zhejiang University
[4]College of Computer Science and Technology, Central South University
[5]State Key Lab of Brain-Machine Intelligence, Zhejiang University

## Abstract

Brain-computer interfaces have shown great potential in motor and speech rehabilitation, but still suffer from low performance stability across days, mostly due to the instabilities in neural signals. These instabilities, partially caused by neuron deaths and electrode shifts, leading to channel-level variabilities among different recording days. Previous studies mostly focused on aligning multi-day neural signals onto a low-dimensional latent manifold to reduce the variabilities, while faced with difficulties when neural signals exhibit significant drift. Here, we propose to learn a channel-level invariant neural representation to address the variabilities in channels across days. It contains a channel-rearrangement module to learn stable representations against electrode shifts, and a channel reconstruction module to handle the missing neurons. The proposed method achieved the state-of-the-art performance with cross-day decoding tasks over two months, on multiple benchmark BCI datasets. The proposed approach showed good generalization ability that can be incorporated to different neural networks.

## 1 Introduction

Brain-computer interfaces (BCIs) have paved a new way for motor and speech rehabilitation (Card et al., 2024; Metzger et al., 2023; Qi et al., 2019). However, the long-term stability of BCIs remains a critical problem due to the channel-wise variability of neural signals. BCI systems commonly require frequent recalibration in use, which hinders the user experience (Ajiboye et al., 2017).

The channel-wise variability can stem from diverse reasons (Degenhart et al., 2020). First, the appearance of new neurons and the disappearance of existing neurons can happen across different experimental days (Figure 1a). These changes affect the distribution of the channel signal, leading to inconsistencies in the input space used for decoding (Flesher et al., 2021; Sussillo et al., 2016). Second, the movement of neurons or electrodes can also drift in the recorded neural signals at each channel (Figure 1b). The situations that cause channel-wise variability can be highly diverse, such that directly learning a channel-wise invariant neural representation can be a challenging problem.

Existing studies addressed this problem from several aspects. The first group is the data alignment approaches, which aim to align the low-dimensional manifolds of neural signals of different days. Researchers have demonstrated that neural signals contain a consistent low-dimensional neural manifold over long periods of time (Gallego et al., 2017). Study (Degenhart et al., 2020) tries to

---

*Corresponding author (qiyu@zju.edu.cn)

39th Conference on Neural Information Processing Systems (NeurIPS 2025).

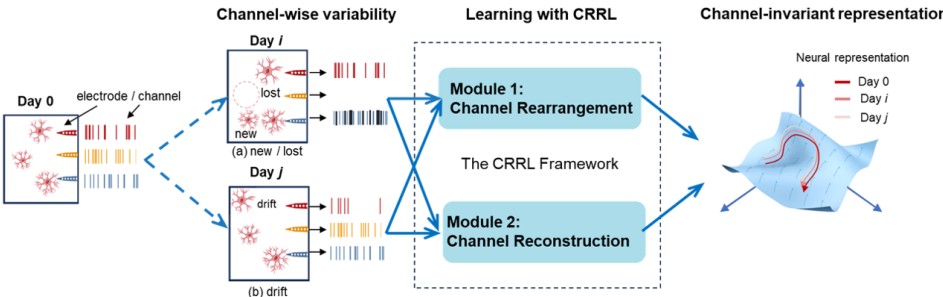

Figure 1: We plot two types of channel-wise variability in neural signals: (a) caused by new or lost neurons. (b) caused by drifted neurons. Our two-module CRRL framework can obtain the channel-invariant representations across days

directly align the manifolds' coordinate axes (the neural recording channels). However, it required that a subset of the channels should be stable across different days. Other studies based on domain adaptation (DA) approaches, which regard neural signals from different days as different domains and attempt to align the signals of these domains, including the ADAN (Farshchian et al., 2018) and Cycle-GAN (Ma et al., 2023) approaches. The second group attempts to improve the robustness of cross-day neural decoding by using more data. NDT2 (Ye et al., 2024) and POYO (Azabou et al., 2024) used a large set of neural data for pretraining to capture the pattern of variation between sessions. These large-scale methods implicitly account for differences in neurons or channels, but require higher training costs. The key challenge is to achieve long-term stable BCI at low cost.

Inspired by variability patterns of the neural signal, we propose a **C**hannel **R**earrangement and **R**econstruction **L**earning framework (CRRL). It contains two main modules: 1) a channel rearrangement module to deal with situations such as new coming neurons explicitly and drifts in neural signals, and 2) a channel reconstruction module to handle situations such as missing neurons. We found that the two operations can highly enhance the channel-level consistency across different days, facilitating a channel-invariant neural representation and improving the neural decoding performance.

We evaluate the proposed method on both simulation datasets with diverse channel-wise variabilities and multiple neural signal datasets with motor, handwriting and speech decoding tasks. With the neural decoding tasks, we evaluate the neural decoding performance of our approach with long-term neural signals with time spans over two months. Experimental results demonstrate that our approach achieves stable decoding performance compared with existing studies, especially for a long period. Moreover, our approach can be incorporated with other methods as a plug-in module, to enhance the neural recording performance and robustness across days.

## 2 Method

The factors causing channel-wise variability can be summarized into two categories: 1) new or lost neurons, and 2) drifted neurons. CRRL uses two modules to handle these two cases separately.

Specifically, in the rearrangement module, we use the permutation matrix obtained by our permutation network to rearrange the signals of day k, and the training objective is to maximize the similarity between each channel on day k and day 0. In the reconstruction module, we employ a channel-based Vector Quantized Variational Autoencoder (VQ-VAE) (Van Den Oord et al., 2017) with randomly masked signals of day 0 to achieve the reconstruction of channel signals and frequency domain features. During the evaluation, we used the data of day k+n as input and obtained the output sequentially through the rearrangement and reconstruction network. Then, we use the output signal to perform downstream decoding tasks, where the decoder is trained by the signals of day 0.

### 2.1 Rearrangement: Channel Permutation Network

Below, we first formulate the tasks of channel permutation and describe the structure of our permutation network. Next, we describe the process of the Sinkhorn-Knopp algorithm (Mena et al., 2018) to

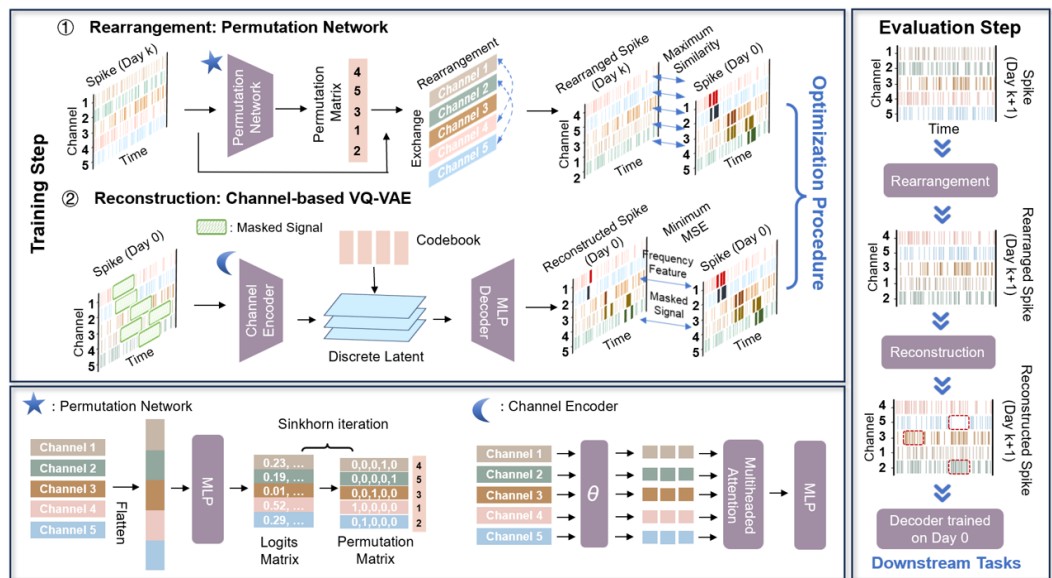

Figure 2: The framework of CRRL. Top: we plot our two-stage training process. Bottom: we plot the network structure of our permutation network and channel encoder. Right: we plot the evaluation process of CRRL, which needs a well-trained permutation network and channel-based VQ-VAE.

obtain differentiable approximations of permutation matrices. Finally, we introduce our loss function for our rearrangement training.

### 2.1.1 Problem Formulation

Given neural signal data from $k$ days $\mathcal{D} = \{\mathbf{X}_0, \mathbf{X}_1, \ldots, \mathbf{X}_{k-1}\}$, where $\mathbf{X}_d \in \mathbb{R}^{N \times C \times T}$ represents the data from day $d$, with $N$ samples, $C$ channels, and $T$ time points, our goal is to find permutation matrices $\{\mathbf{M}_1, \ldots, \mathbf{M}_{k-1}\}$ that match each day's channels with those of day 0. The rearranged data is given by a batched matrix multiplication (BMM) over the time dimension:

$$\mathbf{X}'_d = \mathbf{M}_d^\top \mathbf{X}_d, \quad \text{for } d = 1, \ldots, k-1. \tag{1}$$

### 2.1.2 Predicting Permutation Matrices by Neural Network

To parameterize the permutation matrices in a differentiable manner, we employ a multi-layer perceptron (MLP) $f_\theta$ with shared parameters $\theta$ across all days. We chose the MLP for its flexibility across datasets with different channel counts and for its ability to model arbitrary channel dependencies. For each day $d$, we flatten the data along the channel and time dimensions to form the input to the MLP:

$$\mathbf{x}_d^{\text{flat}} = \text{Flatten}(\mathbf{X}_d) \in \mathbb{R}^{N \times (C \times T)}. \tag{2}$$

The MLP processes the flattened input and reshapes the output to produce logits that represent the unnormalized log probabilities for the permutation matrix $\mathbf{L}_d$:

$$\mathbf{L}_d = f_\theta(\mathbf{x}_d^{\text{flat}}), \quad \mathbf{L}_d \in \mathbb{R}^{N \times C \times C}. \tag{3}$$

### 2.1.3 Sinkhorn Algorithm for Doubly Stochastic Approximation

Permutation matrices are special cases of doubly stochastic matrices with binary entries. To obtain differentiable approximations of permutation matrices while enabling gradient-based optimization, we adopt the Sinkhorn-Knopp algorithm with a continuous relaxation strategy (Mena et al., 2018). This relaxation allows the model to learn permutation patterns in a probabilistic manner during training, while recovering exact permutations via discretization at inference.

**Continuous Relaxation via Gumbel-Sinkhorn**   To bridge the gap between discrete permutations and continuous optimization, we introduce a Gumbel noise perturbation into the logits $\mathbf{L}_d$, following the Gumbel-Sinkhorn framework (Jang et al., 2017a). The perturbed logits are defined as:

$$\tilde{\mathbf{L}}_d = \mathbf{L}_d + \gamma\epsilon, \quad \epsilon \sim \text{Gumbel}(0,1), \tag{4}$$

where $\gamma > 0$ controls the noise intensity. This perturbation encourages exploration during training while maintaining differentiability. And the proof about this differentiability is detailed in Appendix.

The doubly stochastic matrix $\mathbf{M}_d$ is then computed by applying the **Sinkhorn operator** $\mathcal{S}$ to the perturbed logits:

$$\mathbf{M}_d = \mathcal{S}\left(\frac{\tilde{\mathbf{L}}_d}{\tau}\right), \tag{5}$$

where $\tau > 0$ is a temperature parameter. Lower $\tau$ sharpens the output distribution towards a permutation matrix, whereas higher $\tau$ results in a softer distribution.

**Sinkhorn Operator with Iterative Normalization**   The Sinkhorn operator $\mathcal{S}$ iteratively normalizes the input matrix to satisfy row and column stochasticity constraints. Starting from the exponentiated logits:

$$\mathbf{K}_d^{(0)} = \exp\left(\frac{\tilde{\mathbf{L}}_d}{\tau}\right), \tag{6}$$

we perform alternating row and column normalizations for $T_{\text{sink}}$ iterations:

$$\mathbf{K}_d^{(t)} = \mathbf{K}_d^{(t-1)} \oslash \left(\mathbf{K}_d^{(t-1)}\mathbf{1}_C\right) \quad \text{(Row norm)}, \tag{7}$$

$$\mathbf{K}_d^{(t)} = \mathbf{K}_d^{(t)} \oslash \left(\mathbf{1}_C^\top\mathbf{K}_d^{(t)}\right) \quad \text{(Column norm)}, \tag{8}$$

where $\oslash$ denotes element-wise division, and $\mathbf{1}_C$ is a vector of ones. After $T_{\text{sink}}$ iterations, we obtain $\mathbf{M}_d = \mathbf{K}_d^{(T_{\text{sink}})}$.

**Discretization via Hungarian Algorithm at Inference**   During training, $\mathbf{M}_d$ remains a continuous doubly stochastic matrix to preserve differentiability. At inference, we convert $\mathbf{M}_d$ to an exact permutation matrix $\mathbf{P}_d$ using the Hungarian algorithm (Kuhn, 1955):

### 2.1.4   Loss Functions For Rearrangement Training

To train the MLP to produce permutation matrices that effectively align the channels, we define a loss function comprising the negative Pearson correlation loss and an entropy regularization term. Compared to MSE, the negative Pearson correlation loss focuses more on the shape and trend of the signals rather than their absolute amplitudes.

**Negative Pearson Correlation Loss**   For each task $t \in \{1, \ldots, T\}$ and channel $c \in \{1, \ldots, C\}$, we compute the average signal over all samples from day 0 belonging to task $t$:

$$\bar{\mathbf{x}}_{0,t,c} = \frac{1}{N_{0,t}} \sum_{n \in \mathcal{I}_{0,t}} \mathbf{x}_{0,n,c}, \tag{9}$$

where $N_{0,t}$ is the number of samples from day 0 belonging to task $t$, $\mathcal{I}_{0,t}$ is the index set of these samples, $\mathbf{x}_{0,n,c}$ is the signal for sample $n$ and channel $c$ from day 0 data.

Then, the negative Pearson correlation loss is defined as:

$$\mathcal{L}_{\text{corr},d} = -\frac{1}{NC} \sum_{n,c} \frac{\left\langle \mathbf{x}'_{d,n,c} - \overline{\mathbf{x}}'_{d,n,c},\ \bar{\mathbf{x}}_{0,y_n,c} - \overline{\bar{\mathbf{x}}}_{0,y_n,c} \right\rangle}{\left\| \mathbf{x}'_{d,n,c} - \overline{\mathbf{x}}'_{d,n,c} \right\|_2 \left\| \bar{\mathbf{x}}_{0,y_n,c} - \overline{\bar{\mathbf{x}}}_{0,y_n,c} \right\|_2}, \tag{10}$$

where $\mathbf{x}'_{d,n,c}$ is the signal for sample $n$ and channel $c$ from reordered day $d$ data, $y_n$ is the task label of sample $n$, $\bar{\mathbf{x}}_{0,y_n,c}$ is the average signal over all day 0 samples belonging to task $y_n$ and channel $c$, $\overline{\mathbf{x}}'_{d,n,c}$ and $\overline{\bar{\mathbf{x}}}_{0,y_n,c}$ are the mean over time of $\mathbf{x}'_{d,n,c}$ and $\bar{\mathbf{x}}_{0,y_n,c}$, respectively.

The total correlation loss is then:

$$\mathcal{L}_{\text{corr}} = \sum_{d=1}^{k-1} \mathcal{L}_{\text{corr},d}. \tag{11}$$

**Entropy Regularization**    To encourage the approximation matrices $\mathbf{M}_d$ to be close to true permutation matrices, we add an entropy regularization term:

$$\mathcal{L}_{\text{entropy},d} = -\frac{1}{N} \sum_{n}^{N} \sum_{i}^{C} \sum_{j}^{C} M_{d,n,i,j} \log M_{d,n,i,j}. \tag{12}$$

The total entropy regularization loss is:

$$\mathcal{L}_{\text{entropy}} = \sum_{d=1}^{k-1} \mathcal{L}_{\text{entropy},d}. \tag{13}$$

**Total Loss for Stage One**    The overall loss function for stage one is:

$$\mathcal{L}_{\text{stage1}} = \mathcal{L}_{\text{corr}} + \lambda_{\text{entropy}} \mathcal{L}_{\text{entropy}}. \tag{14}$$

The two components of the loss function guarantee that the rearranged signal on day k closely resembles that of day 0 while also optimizing the approximation of the permutation matrix. After training, the parameters of the permutation network are fixed for the subsequent training or evaluation phase.

## 2.2    Reconstruction: Channel-based VQ-VAE

Firstly, we formulate the tasks of channel reconstruction. Then, we describe the architecture of our channel-based VQ-VAE. VQ-VAE discretizes the continuous latent space using vector quantization, creating a discrete codebook which is suitable for data with discrete characteristics, such as spike signals. Discrete encoding enhances robustness to input noise and effectively handles random disturbances in spike signals. Next, we introduce the loss function of our reconstruction training.

### 2.2.1    Problem Formulation

Given neural signal data from day 0 and the rearranged data from day $k$, which are mixed together as $X_{mix}$ in a ratio of 7:3. Our goal is to reconstruct missing or corrupted segments in the neural signals. During training, we simulate such conditions by randomly masking a signal segment from one of the channels in each sample. Specifically, for each sample $n$, we randomly select a channel $c_n$ and a time interval $[ti_{\text{start}}, ti_{\text{end}}]$, and set:

$$\tilde{\mathbf{x}}_{mix,n,c_n}[ti_{\text{start}} : ti_{\text{end}}] = 0, \tag{15}$$

where $\tilde{\mathbf{x}}_{mix,n,c_n}$ is the masked signal for sample $n$ and channel $c_n$.

### 2.2.2    VQ-VAE Architecture

The architecture of the VQ-VAE is illustrated in Figure 2. It consists of an encoder that independently embeds the raw time series from different channels as tokens, a cross-channel attention module that captures inter-channel dependencies, and a decoder comprising an MLP network that reconstructs the signals. The VQ-VAE also includes a vector quantization layer for discretizing the latent representations.

**Channel Encoder**   The encoder $E_\phi$ processes the masked signals to produce latent representations.

For each sample $n$, channel $c$ and time $Ti$, the raw signal $\tilde{\mathbf{x}}_{mix,n,c} \in \mathbb{R}^{Ti}$ is embedded using a fully connected layer to produce a token representation:

$$\mathbf{e}_{n,c} = \text{Embed}(\tilde{\mathbf{x}}_{mix,n,c}) = \mathbf{W}_e \tilde{\mathbf{x}}_{mix,n,c} + \mathbf{b}_e \in \mathbb{R}^{d_e}, \tag{16}$$

where $\mathbf{W}_e \in \mathbb{R}^{d_e \times T}$ and $\mathbf{b}_e \in \mathbb{R}^{d_e}$ are learnable parameters, and $d_e$ is the embedding dimension.

The embeddings from all channels are input into a cross-channel attention mechanism to capture dependencies among channels. We stack the embeddings for each sample to form $\mathbf{E}_n = [\mathbf{e}_{n,1}; \mathbf{e}_{n,2}; \ldots; \mathbf{e}_{n,C}] \in \mathbb{R}^{C \times d_e}$.

We apply a self-attention mechanism (Vaswani, 2017):

$$\mathbf{H}_n = \text{MultiHeadAttention}(\mathbf{E}_n), \tag{17}$$

where $\mathbf{H}_n \in \mathbb{R}^{C \times d_e}$ contains the attention representations for each channel.

Each attention representation $\mathbf{h}_{n,c} \in \mathbb{R}^{d_e}$ is passed through a multi-layer perceptron to obtain the latent representation:

$$\mathbf{z}_{n,c} = \text{ReLU}\left(\mathbf{h}_{n,c}\mathbf{W}_1 + \mathbf{b}_1\right)\mathbf{W}_2 + \mathbf{b}_2 \in \mathbb{R}^d, \tag{18}$$

where $\mathbf{W}_1 \in \mathbb{R}^{d_e \times d_{\text{ff}}}$, $\mathbf{W}_2 \in \mathbb{R}^{d_{\text{ff}} \times d}$, and $\mathbf{b}_1$, $\mathbf{b}_2$ are biases.

**Vector Quantization Layer**   We use a codebook $\mathcal{C} = \{\mathbf{e}_k\}_{k=1}^K$, where each codeword $\mathbf{e}_k \in \mathbb{R}^d$. Each latent vector $\mathbf{z}_{n,c}$ is quantized by finding the nearest codeword:

$$\mathbf{q}_{n,c} = \mathbf{e}_{k^*}, \quad \text{where } k^* = \arg\min_k \|\mathbf{z}_{n,c} - \mathbf{e}_k\|_2^2. \tag{19}$$

**MLP Decoder**   The decoder $D_\psi$ reconstructs the signals from the quantized embeddings. It consists of a multi-layer perceptron:

$$\hat{\mathbf{x}}_{n,c} = \mathbf{W}_d\mathbf{q}_{n,c} + \mathbf{b}_d \in \mathbb{R}^T, \tag{20}$$

where $\mathbf{W}_d \in \mathbb{R}^{T \times d}$ and $\mathbf{b}_d \in \mathbb{R}^T$ are learnable parameters.

### 2.2.3   Loss Function For Reconstruction Training

We train the VQ-VAE by minimizing a loss function, including reconstruction and commitment losses. We reconstruct signals in both the time domain and the frequency domain simultaneously to improve the stability of the reconstruction.

**Time Domain Reconstruction Loss**   We compute the mean squared error (MSE) between the reconstructed signals and the original signals for the masked entries $m_{n,c}$:

$$\mathcal{L}_{\text{time}} = \frac{1}{\sum_{n,c} m_{n,c}} \sum_{n,c} m_{n,c} \|\hat{\mathbf{x}}_{n,c} - \mathbf{x}_{n,c}\|_2^2. \tag{21}$$

**Frequency Domain Reconstruction Loss**   We also compute the MSE in the frequency domain features (amplitude $\mathcal{A}(\cdot)$ and phase $\phi(\cdot)$ via Discrete Fourier Transform (DFT)):

$$\mathcal{L}_{\mathcal{A}} = \frac{1}{\sum m_{n,c}} \sum_{n,c} m_{n,c} \|\mathcal{A}(\hat{\mathbf{x}}_{n,c}) - \mathcal{A}(\mathbf{x}_{n,c})\|_2^2 \tag{22}$$

$$\mathcal{L}_{\phi} = \frac{1}{\sum m_{n,c}} \sum_{n,c} m_{n,c} \|\phi(\hat{\mathbf{x}}_{n,c}) - \phi(\mathbf{x}_{n,c})\|_2^2 \tag{23}$$

$$\mathcal{L}_{\text{freq}} = \mathcal{L}_{\mathcal{A}} + \mathcal{L}_{\phi} \tag{24}$$

**Vector Quantization Commitment Loss**   The commitment loss encourages the encoder outputs to be close to the codewords:

$$\mathcal{L}_{\text{vq}} = \frac{1}{NC} \sum_{n,c} \left\| \text{sg}[\mathbf{z}_{n,c}] - \mathbf{q}_{n,c} \right\|_2^2 + \beta \left\| \mathbf{z}_{n,c} - \text{sg}[\mathbf{q}_{n,c}] \right\|_2^2, \tag{25}$$

where $\text{sg}[\cdot]$ denotes the stop-gradient operator, and $\beta$ is the commitment cost weight.

**Total Loss**   The total loss function is:

$$\mathcal{L}_{\text{stage2}} = \mathcal{L}_{\text{time}} + \lambda_{\text{freq}} \mathcal{L}_{\text{freq}} + \lambda_{\text{vq}} \mathcal{L}_{\text{vq}}, \tag{26}$$

where $\lambda_{\text{freq}}$ and $\lambda_{\text{vq}}$ are hyperparameters.

Figure 2 illustrates the conceptual CRRL training and evaluation processes. Algorithms 1 and 2 specify the two stages of CRRL training in detail. In the rearrangement training, the first step is to predict the permutation matrices. Then, CRRL rearranges the spike channels and obtains the rearranged spike. In the reconstruction training, we predict the masked signal and its frequency feature. During optimization, our approach learns to robustly restore the information of the day 0 signal. For evaluation, we can obtain the reconstructed data by well-trained CRRL. The obtained data can decode downstream tasks by a decoder trained on day 0.

## 3   Experiments

### 3.1   Setup

We evaluate our method for experiments on simulation and real neural datasets. For the simulation dataset, we simulate two channel-wise variability in the simulation dataset: 1) new or lost neurons and 2) drifted neurons. For 1), we delete/add part of the input channels. For 2), we randomly shuffle the input channels. Additionally, we set different change ratios (5 % or 10 %) to compare the impact of different ratios on performance. To simulate the plasticity of neurons, we also replace part of the neurons with some reserved neurons that have different tuning curves. For real datasets, we use two human datasets and five monkey datasets. The human datasets include a handwriting dataset (Willett et al., 2021) and a speech dataset (Willett et al., 2023). And the monkey datasets (Dyer et al., 2017) comprise two Center-out task datasets ((C) and (M)), two isometric wrist task datasets ((J) and (S)), and a key grasping task dataset (G).

To evaluate and compare the performance of models in different periods, we divided all the datasets into five different time intervals according to the number of days from day 0. The dataset and split details are described in Appendix A.

**Baseline Methods.**   We compared our method with five cross-day decoding methods used in BCI. 1) A manifold-based method Stabilizedbci (Degenhart et al., 2020); 2) Another manifold-based method NoMAD (Karpowicz et al., 2022); 3) A domain adaptation method SD-Net (Fang et al., 2023); 4) An adversarial domain adaptation method ADAN (Farshchian et al., 2018); 5) Another adversarial domain adaptation method Cycle-GAN (Ma et al., 2023). The detailed hyper-parameter settings of these methods are described in Appendix B.

We used SVM to perform classification tasks for decoding in downstream tasks (Qi et al., 2022, 2025) and Wiener Filter (Chen et al., 2006) to perform regression tasks.

### 3.2   Results on Simulation Dataset

We evaluate CRRL under three simulated neural instabilities (Table 1, 11). We ablate either the Rearrangement Module (RA) or Reconstruction Module (RC) to analyze module contributions. Here, we use Coefficient of Determination ($R^2$) and accuracy (acc) to evaluate the performance of regression and classification on the simulation dataset, respectively.

Table 1 reported that using two modules together gives the best performance in all scenarios. For the scenario of new/lost neurons, without RC (w/o RC) degrades performance significantly (0.92/99.5% to

Table 1: Ablation study on simulation dataset.

| $R^2/acc$ | w/o RA | w/o RC | RA + RC |
|---|---|---|---|
| New/lost neuron (5 %) | 0.83/99.1 | 0.72/92.7 | **0.92/99.5** |
| New/lost neuron (10 %) | 0.81/96.1 | 0.69/89.3 | **0.87/99.3** |
| Shuffled channel (5 %) | 0.15/55.1 | 0.85/92.1 | **0.89/98.0** |
| Shuffled channel (10 %) | 0.05/48.3 | 0.77/87.5 | **0.86/98.2** |
| Changed function (5 %) | 0.66/80.8 | 0.45/64.7 | **0.73/91.4** |

Table 2: The performance of regression and classification prediction which compares with different cross-day decoding methods.

| $R^2/acc$ | Model | Task | Number of days since day 0 | | | | | |
|---|---|---|---|---|---|---|---|---|
| | | | 0 | [5, 10) | [10, 20) | [20, 40) | [40, 65) | [65, 100) |
| Center-Out (C) | Stabilizedbci | Trajectory/ Direction | 0.84/86.5 | 0.59/56.6 | 0.52/50.7 | 0.45/44.8 | -/- | -/- |
| | SD-Net | | | 0.73/69.3 | 0.65/62.5 | 0.62/63.3 | -/- | -/- |
| | NoMAD | | | 0.55/48.4 | 0.58/50.6 | 0.41/37.9 | -/- | -/- |
| | **CRRL (Ours)** | | | **0.75/77.1** | **0.68/70.5** | **0.65/66.8** | -/- | -/- |
| Center-Out (M) | Stabilizedbci | Trajectory/ Direction | 0.68/73.1 | -/- | 0.38/51.6 | 0.34/43.7 | -/- | -/- |
| | SD-Net | | | -/- | 0.45/62.0 | 0.39/52.0 | -/- | -/- |
| | NoMAD | | | -/- | 0.33/39.5 | 0.26/29.7 | -/- | -/- |
| | **CRRL (Ours)** | | | -/- | **0.51/63.5** | **0.45/57.4** | -/- | -/- |
| ISO (J) | Stabilizedbci | Trajectory/ Direction | 0.73/92.5 | 0.38/57.4 | 0.32/53.1 | 0.23/35.0 | 0.20/36.5 | 0.17/29.1 |
| | SD-Net | | | 0.42/65.2 | 0.35/58.6 | 0.28/44.2 | 0.25/43.8 | 0.20/34.5 |
| | NoMAD | | | 0.44/65.2 | 0.45/**62.4** | 0.37/40.8 | 0.28/37.5 | 0.31/39.7 |
| | **CRRL (Ours)** | | | **0.51/66.3** | **0.48**/61.5 | **0.43/62.2** | **0.46/48.6** | **0.47/51.1** |
| ISO (S) | Stabilizedbci | Trajectory/ Direction | 0.75/94.4 | 0.33/43.3 | 0.35/47.2 | 0.24/40.9 | 0.18/25.0 | <0 /14.8 |
| | SD-Net | | | 0.40/56.5 | 0.37/53.7 | 0.36/53.7 | 0.32/45.6 | 0.24/35.9 |
| | NoMAD | | | 0.34/53.3 | **0.40**/51.2 | **0.38**/45.0 | 0.37/44.1 | 0.29/31.3 |
| | **CRRL (Ours)** | | | **0.43/57.2** | 0.39/**56.6** | 0.35/**54.5** | **0.37/54.1** | **0.30/45.2** |
| ISO (J) | ADAN | EMG | 0.71 | 0.66 | 0.60 | 0.56 | 0.54 | 0.48 |
| | Cycle-GAN | | | 0.68 | 0.65 | 0.63 | 0.59 | 0.50 |
| | **CRRL (Ours)** | | | **0.70** | **0.66** | **0.69** | **0.65** | **0.61** |
| Key (G) | ADAN | EMG | 0.46 | 0.26 | 0.21 | 0.18 | <0 | - |
| | Cycle-GAN | | | 0.27 | 0.25 | 0.20 | 0.09 | - |
| | **CRRL (Ours)** | | | **0.41** | **0.37** | **0.35** | **0.32** | - |

0.72/92.7%), indicating RC's critical role in compensating for neuron loss. For the scenario of drifted neurons, w/o RA gives the worst performance, and w/o RC achieves the second-best performance, suggesting RA is vital for handling neuron drifting. When tuning function changes, RA+RC achieves 91.4% accuracy with a 26.7% relative improvement over the w/o RC. RC contributes more to $R^2$ and acc (0.66/80.8% compared to 0.45/64.7%), implying it effectively adapts to latent representational shifts.

### 3.3 Results on Speech and Handwriting Decoding

We compare our method with SD-Net and NoMAD on speech and handwriting datasets. The decoding target of the speech dataset is 7 words with a "do nothing" condition, where we use acc as the evaluation metric. Similarly, the decoding target of the handwriting dataset is 31 characters, and we also use acc as the metric.

Figure 3 shows CRRL's improvements in cross-day decoding performance with SD-Net and NoMAD. For the speech dataset, CRRL shows consistent performance on different days. Notably, CRRL achieves accuracy recovery to 86% on day 21 while accuracies of SD-Net and NoMAD decline to 62% and 42% respectively. Moreover, the performance advantage of CRRL is more obvious when the time span is longer.

For the handwriting dataset, Our CRRL framework maintains superior long-term performance, outperforming existing methods significantly (the pairwise statistical tests are described in Appendix B). CRRL shows stability with 67% accuracy at Day 16 (vs SD-Net's 57% and NoMAD's 32%).

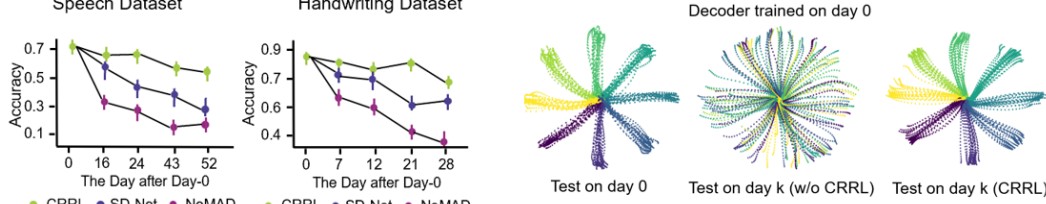

Figure 3: (Left) we compare our CRRL with two baseline methods (SD-Net and NoMAD) on speech and handwriting datasets. We evaluate the performance using the classifier trained on day 0 data. (Right) we use CRRL to decode the hand trajectory across days. First, we use the regressor trained with the data from day 0 to evaluate the performance. In the middle figure, this regressor completely loses its decoding ability on day k data. However, through CRRL, the regressor recovers its performance, which is close to the performance on day 0 data.

Table 3: Ablation Study on Real Neural Datasets.

| | Component | | Center-Out (C) | | | ISO (J) | | | Key (G) | | |
|---|---|---|---|---|---|---|---|---|---|---|---|
| | RA | RC | [5, 10) | [10, 20) | [20, 40) | [5, 10) | [10, 20) | [20, 40) | [5, 10) | [10, 20) | [20, 40) |
| CRRL | ✓ | - | 0.73/68.1 | 0.61/64.6 | 0.57/63.5 | 0.44/57.7 | 0.41/48.3 | 0.41/52.5 | 0.35 | 0.31 | 0.32 |
| CRRL | - | ✓ | 0.73/66.4 | 0.64/57.6 | 0.52/53.2 | 0.47/62.2 | 0.36/53.0 | 0.32/49.3 | 0.38 | 0.33 | 0.26 |
| CRRL | ✓ | ✓ | **0.75/77.1** | **0.68/70.5** | **0.65/66.8** | **0.51/66.3** | **0.48/61.5** | **0.43/62.2** | **0.41** | **0.37** | **0.35** |

## 3.4 Results on Motor Decoding

We compare CRRL with Stabilizedbci, SD-Net, and NoMAD on five monkey neural datasets, which include diverse motor decoding tasks. The total number of sessions (which is equal to the total number of days) used for each dataset. For hand trajectory and electromyography (EMG) decoding, we use $R^2$ as an evaluation metric. For the eight-direction classification, we use accuracy as a metric.

Table 2,8 shows that CRRL achieves consistent superiority in both trajectory regression ($R^2$) and direction classification (acc) tasks, particularly under challenging long-term decoding scenarios. For the Center-Out datasets (C and M), CRRL maintains stable performance as days progress. On the Center-Out (C), CRRL achieves $R^2$/acc of 0.75/77.1% for days [5, 10], surpassing SD-Net (0.73/69.3%) and Stabilizedbci (0.59/56.6%). Notably, even after 20–40 days, CRRL retains 0.65/66.8, exhibiting only a 12.8% drop in $R^2$ compared to SD-Net's 15.1% decline. On the ISO (J) dataset, CRRL achieves 0.47/51.1% $R^2$/acc at [65, 100] days, outperforming NoMAD (0.31/39.7%) and SD-Net (0.20/34.5%). For the challenging Key (G) dataset, CRRL maintains 0.35 $R^2$ at [20, 40] days, a 94% improvement over Cycle-GAN (0.20).

## 3.5 Ablation Study

We conduct ablation study experiments on Center-Out (C), ISO (J) and Key (G) datasets to investigate the effectiveness of two modules of CRRL (shown in Table 3). The decoding target of ISO (J) is hand trajectory. On the Center-Out (C) dataset, RA + RC achieves 0.75 $R^2$ in [5,10) days, which is 2.7% higher than RA-alone (0.73) and RC-alone (0.73). This advantage of performance grew over time, delivering 12.3% accuracy improvement (66.8% vs 63.5%/53.2%) at [20,40) days. Results on the ISO (J) dataset demonstrate that RA + RC achieves classification accuracy of 62.2% at [20,40) days, outperforming individual components by 18.7% (RC) and 18.4% (RA). Notably, RC proves particularly crucial for Key (G) EMG decoding, where RA achieves 0.38 accuracy at [5,10) days vs RA's 0.35. However, RA + RC yields 7.9% additional gain (0.41).

## 3.6 Comparison with Large-Scale Models

The goal of our method is to achieve long-term stable BCI decoding using a small amount of data for training. As shown in Table 4, CRRL w/o PT, which only uses 5 days of data for training, achieves a performance close to large-scale methods such as POYO-1 (Azabou et al., 2024) and NDT2 Multi (Ye et al., 2024).

Table 4: Comparison with large-scale models including POYO-1 and NDT2 Multi. PT is the pretrain model and FSS is the few shot setting,

| Method | Monkey C | M1-A | M2 |
|---|---|---|---|
| POYO-1 + Full finetune | $0.9683 \pm 0.01$ | - | - |
| NDT2 Multi + FSS | - | $0.59 \pm 0.07$ | $0.43 \pm 0.08$ |
| CRRL w/o PT | $0.8715 \pm 0.03$ | $0.42 \pm 0.13$ | $0.31 \pm 0.07$ |
| **CRRL PT + Full finetune** | **$0.9750 \pm 0.02$** | **$0.63 \pm 0.08$** | **$0.55 \pm 0.13$** |

Table 5: Plugin-in experiment.

| $R^2$ | Day 7 | Day 30 | Day 56 |
|---|---|---|---|
| ADAN | $0.26 \pm 0.04$ | $0.18 \pm 0.05$ | <0 |
| Cycle-GAN | $0.27 \pm 0.02$ | $0.20 \pm 0.03$ | $0.09 \pm 0.02$ |
| RA + ADAN | $0.29 \pm 0.02$ | $0.23 \pm 0.03$ | $0.14 \pm 0.03$ |
| **RA + Cycle-GAN** | **$0.33 \pm 0.01$** | **$0.27 \pm 0.01$** | **$0.20 \pm 0.02$** |

For a fully comparison, we perform large-scale pretraining using more training data. The pre-training method is conducted as multiple full training runs on different monkeys, but all runs share the same RA and RC module. Specifically, we adopt a sequential training strategy: the RA module is first trained on one monkey's dataset (e.g., Monkey M), and then its learned parameters are used to initialize training on the next monkey, and so on. (Since two sessions of Monkey C were selected as the test set when compared with POYO, the final training was performed on Monkey C). Each training run aligns Day k to Day 0 within the same subject. After the RA module is trained on a given monkey, we immediately train the corresponding RC module to reconstruct Day 0 neural activity from the RA-aligned Day k input.

This progressive training enables the RA and RC modules to accumulate cross-subject channel alignment knowledge while maintaining subject-specific alignment through Day 0 supervision. To mitigate potential forgetting during this sequential training process, we adopt a lightweight replay strategy: when training on a new monkey, we add a small number of samples (10%) from previously seen monkeys in the training set. As shown in Table 4, the setting of CRRL PT + Full finetune achieves a better performance compared to POYO-1 and NDT2.

### 3.7 Combining CRRL Module into DA Methods

Since our RC module can be regarded as a special domain adaptation method, and RA is a pre-alignment before adaptation, it can be incorporated with the existing DA method as a plug-in. We evaluate the performance of plug-ins on the Key (G) dataset. As demonstrated in Table 5. When combined with Cycle-GAN, the RA module boosts $R^2$ scores by 22.2% on Day 7 (0.33 vs 0.27 baseline) and maintains 122% higher accuracy by Day 56 (0.20 vs 0.09). Similar improvements are shown in ADAN. Notably, the RA+Cycle-GAN achieves better performance separation over time: 6.7% advantage on Day 7 expands to 11.1% by Day 30 and 11.0% on Day 56. These results suggest CRRL's effectiveness design enables plug-and-play enhancement of DA methods.

## 4 Conclusion

In this work, we propose CRRL, a Channel Rearrangement and Reconstruction Learning framework. Our model is inspired by the channel-wise variability in neural signals. By adapting different types of variability in the model, it effectively alleviates the damage of this variability on the decoder and achieves state-of-the-art performance on multiple tasks. CRRL has good robustness, which makes its advantage more obvious when the time span is long. In addition, CRRL can also be combined with other methods as plug-ins to improve the performance of other methods.

However, although our work achieves better performance than the existing methods, we still observe a slow decline in performance over time, which may be due to slow changes in neuronal population modulation caused by learning or environmental changes. How to accurately capture the modulation change patterns of neuronal populations is another interesting direction for future work. Overall, we hope the proposed cross-day decoding framework will inspire new intelligence paradigms and provide a tool for long-term stable clinical BCIs.

## 5 Acknowledgement

This work was supported by grants from the Key Research and Development Program of Zhejiang Province in China (2023C03001), the Natural Science Foundation of China (62276228), and the Zhejiang Provincial Natural Science Foundation (LR24F020002).

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

# A  Experiments Details

## A.1  Simulation Dataset Details

We conducted the simulation dataset based on the cosine tuning curve model (Gilja et al., 2012): The tuning function links motor cortex signals to hand movement direction. This biological tuning can be simulated using a cosine function:

$$f_i(t) = a_{i,0} + a_{i,x}cos\theta_t + a_{i,y}sin\theta_t + \epsilon_i, \tag{27}$$

where $f$ denotes the estimated firing rates, $a_{i,0}$, $a_{i,x}$, $a_{i,y}$ are the coeffcients including baseline, $cosPD_i$ and $sinPD_i$ respectively, PD means the preferred direction, the $\epsilon$ means the noise term and $\theta_t$ means the moving direction at time $t$.

We simulate an 8-direction center-out reaching task, and each trial corresponds to a reach toward one of 8 radial targets. We evaluate two tasks: 1) Classification: Predicting the intended reach direction (8 classes). 2) Regression: Reconstructing 2D hand velocity at each timepoint.

The simulation dataset consists of: 1) 100 neurons, 2) 50 timepoints per trial, 3) 800 trials total (100 per direction).

## A.2  Real Dataset Details

In Section 3, we have described the datasets used in the experiments (Dyer et al., 2017; Willett et al., 2023, 2021). Here, we give their details and download links:

- **Isometric wrist (ISO)** .[2] In the Isometric wrist datasets, Monkeys J and S performed an isometric wrist task, controlling a cursor by applying forces on a padded box. Each trial began with a center target hold (0.2–1.0 s), followed by a randomly selected outer target and an auditory cue. EMG data were collected from 7 muscles in Monkey J's left arm (ECU, FCU, ECRl, EDC, FCR, FDP, ECRb) and 8 muscles in Monkey S's left arm and hand (FDP1, FDP2, FCR, 1DI, FPB, EDC, MD, ECRl). Trial lengths varied as no temporal alignment was performed.

- **Center-out reach**.[2] In the Center-Out datasets, Monkeys C and M performed a center-out reaching task using a planar manipulandum, moving the handle in a parasagittal plane. Each trial began with the monkey moving to the center of the workspace, followed by a random waiting period before one of eight outer targets appeared. Trials were extracted from the 'go cue time' to the 'trial end', as the monkeys remained stationary before the go cue.

- **Key Grasping**.[2] In Key Grasping dataset, Monkey G performed a grasping task, reaching and grasping a small rectangular cuboid under the screen using a key grasp with the thumb and index finger. Each trial began with the monkey resting its hand on a touchpad for a random period (0.5–1.0 s). Trials were extracted from the 'go cue time' to the 'trial end', as movements were random before the go cue. EMG data were collected from 8 muscles in the left arm and hand (FCR, FDP, PT, FPB, 1DI, SUP, ECU, EDC).

- **Handwriting**.[3] In the handwritten dataset, the neural activity was recorded from two microelectrode arrays which including 192 channels in the motor area. In the experiments of handwriting dataset, the participant attempted to write 31 different characters in response to cues shown on a computer screen. The cue list includes the following characters:'a', 'b', 'c', 'd', 'e', 'f', 'g', 'h', 'i', 'j', 'k', 'l', 'm', 'n', 'o', 'p', 'q', 'r', 's', 't', 'u', 'v', 'w', 'x', 'y', 'z', 'greaterThan', 'comma', 'apostrophe', 'tilde', 'questionMark'.

- **Speech**.[4] In the speech dataset, the neural activity was recorded from four microelectrode arrays (each array includes 64 channels and we only use the first two arrays here). In the experiments of speech dataset, the participant attempted to speak 7 single words (and a "do nothing" condition) in response to cues shown on a computer. The cue list includes the following words: 'choice', 'day', 'kite', 'though', 'veto', and 'were'.

---

[2] https://datadryad.org/stash/dataset/doi:10.5061/dryad.cvdncjt7n
[3] https://datadryad.org/stash/dataset/doi:10.5061/dryad.wh70rxwmv
[4] https://datadryad.org/stash/dataset/doi:10.5061/dryad.x69p8czpq

For every category, we used an equal number of training samples, resulting in a balanced number of trials across all classes. More parameters of these datasets are shown in Table 6.

Table 6: Detailed dataset parameters.

| Dataset | Range of days | Num of recordings | Day 0 | Decoding target | Decoding Task |
|---|---|---|---|---|---|
| ISO (J) | 95 | 20 | 20150730 | EMGs, hand trajectory and 8 directions | Regression & Classification |
| ISO (S) | 83 | 18 | 20120821 | EMGs, hand trajectory and 8 directions | Regression & Classification |
| Center-Out (C) | 38 | 12 | 20160927 | Hand trajectory and 8 directions | Regression & Classification |
| Center-Out (M) | 32 | 11 | 20140203 | Hand trajectory and 8 directions | Regression & Classification |
| Key (G) | 53 | 8 | 20190812 | EMGs | Regression |
| Handwriting | 51 | 10 | 20191125 | 31 characters | Classification |
| Speech | 41 | 9 | 20220616 | 7 words and a "do nothing" condition | Classification |

## A.3 Dataset usage of Section 3.6

In Section 3.6, we compare CRRL with large-scale methods such as POYO and NDT-2, which can implicitly account for differences in neurons or channels (in the multiunit recording case) across days through learnt embeddings. We added two variants: CRRL w/o PT and CRRL PT + Full finetune. The former does not perform additional pre-training, while the latter uses the data that was used in the comparison methods for pre-training and finetuning.

For comparison with POYO-1, we use datasets from seven non-human primates spanning three different labs, with a total of 27,373 units and 16,473 neuron-hours for training and finetuning. And we use the two held-out sessions from Monkey C (CO task) for testing. The standard deviation is reported over the sessions.

For comparison with NDT-2, we use datasets M1-A, which consists of 4 held-in datasets spanning 5 days, each with 53-61 minutes of calibration data, and 3 held-out datasets spanning 21 days, which have only 1.1-2.2 minutes of calibration data available. And M2 includes 4 held-in datasets over 10 days, with between 5.9-13.3 minutes of calibration data per session available, and 4 held out datasets over 26 days with between 0.8-1.7 minutes of calibration data provided. The FSS (Few Shot Supervised) setting means that it uses a few calibration data in held-out datasets for f inetuning.

## A.4 Dataset Split

To investigate the decoder's performance changes over different time periods, we divide all the datasets into five time periods: [5, 10], [10, 20), [20, 40), [40, 65), [65, 100). The data before day 5 includes day 0 as training data (Figure 4). In particular, in the handwriting and Center-Out (M) datasets, there is no data before day 5, and we take the training set to the first day after day 0. For each time period, we compute the average of all data performance in the time period. We average performance across 10 random seeds for each data point. In the cross-day setting, we use data from the first 5 consecutive days for training (held-in data). All data from these 5 days are merged, then randomly split into 90% training and 10% validation. Data from Day 6 and onward is treated as held-out test data. The model is evaluated on these later days without any additional training or adaptation, to assess generalization under temporal distribution shift. And the data split ratio used for training the day 0 decoder of all datasets is 8:1:1.

## A.5 Metric Details

We adopt the **Accuracy (ACC)** metric to evaluate classification tasks (8 directions, phonemes and characters), which measures the proportion of correctly predicted samples. For regression tasks (hand trajectory), we use the **Coefficient of Determination ($R^2$)** to assess the proportion of variance in the target variable explained by the model. In multivariate regression tasks (EMGs), we extend $R^2$ to **Multi-Target $R^2$** to evaluate the model's performance across multiple output variables simultaneously. These metrics are calculated as follows:

**Accuracy (ACC):**

$$\text{ACC} = \frac{1}{n} \sum_{i=1}^{n} \mathbb{I}(y_i = \hat{y}_i) \tag{28}$$

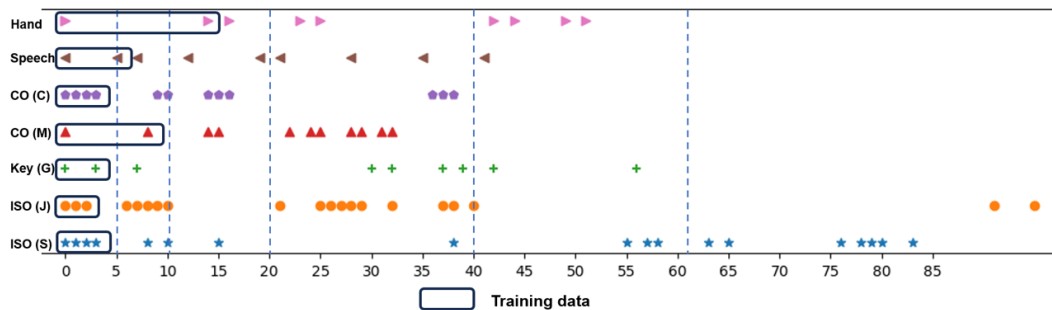

Figure 4: We show the selection of training data for different datasets (black rectangle). Data from day 0-5 were selected as training data for most of the datasets. We divide the different time phases with blue dashed lines, which correspond to the time phases of Table 2, 7.

where $y_i$ represents the true label, $\hat{y}_i$ represents the predicted label, $\mathbb{I}(\cdot)$ is the indicator function, and $n$ is the total number of samples.

**Coefficient of Determination ($R^2$)**:

$$R^2 = 1 - \frac{\sum_{i=1}^{n}(y_i - \hat{y}_i)^2}{\sum_{i=1}^{n}(y_i - \bar{y})^2} \tag{29}$$

where $y_i$ represents the actual value, $\hat{y}_i$ represents the predicted value, and $\bar{y}$ is the mean of the actual values.

**Multi-Variate $R^2$**:

$$\text{Multi-Variate } R^2 = 1 - \frac{\sum_{j=1}^{m}\sum_{i=1}^{n}(y_{i,j} - \hat{y}_{i,j})^2}{\sum_{j=1}^{m}\sum_{i=1}^{n}(y_{i,j} - \bar{y}_j)^2} \tag{30}$$

where $y_{i,j}$ represents the actual value of the $j$-th target for the $i$-th sample, $\hat{y}_{i,j}$ represents the predicted value of the $j$-th target for the $i$-th sample, $\bar{y}_j$ is the mean of the actual values for the $j$-th target, $m$ is the number of targets, and $n$ is the number of samples.

These metrics collectively offer a comprehensive evaluation of model performance across diverse tasks.

## B    Additional Experiments

The results of main experiment with error intervals (the standard deviation) are as follow:

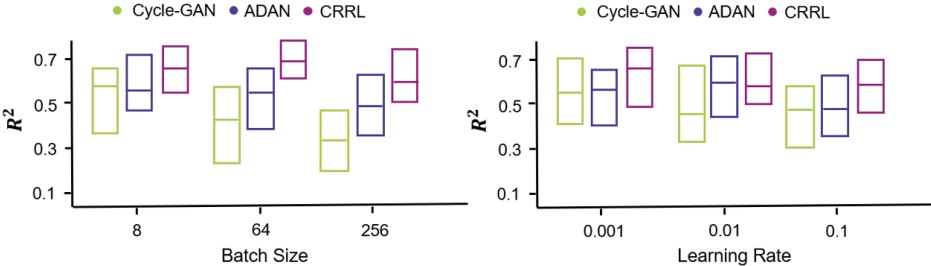

Figure 5: Comparison on different parameter setting.

### B.1    Analysis of Parameter Sensitivity

We compare the parameter sensitivity of our method with Cycle-GAN and ADAN. In the experiment, we selected a recording in the ISO (J) dataset (Day 30), and the decoding target was EMG. We compare the performance of the models with different batch sizes (BS) and learning rates (LR),

Table 7: The performance of regression and classification prediction which compares with different cross-day decoding methods.

| $R^2/acc$ | Model | Task | Number of days since day 0 | | | | | |
|---|---|---|---|---|---|---|---|---|
| | | | 0 | [5, 10) | [10, 20) | [20, 40) | [40, 65) | [65, 100) |
| Center-Out (C) | Stabilizedbci | Direction | 86.5± 0.011 | 56.6± 0.027 | 50.7± 0.157 | 44.8± 0.110 | -/- | -/- |
| | SD-Net | | | 69.3± 0.152 | 62.5± 0.005 | 63.3± 0.232 | -/- | -/- |
| | NoMAD | | | 48.4± 0.023 | 50.6± 0.157 | 37.9± 0.226 | -/- | -/- |
| | **CRRL (Ours)** | | | **77.1± 0.117** | **70.5± 0.198** | **66.8± 0.117** | -/- | -/- |
| Center-Out (M) | Stabilizedbci | Direction | 73.1± 0.208 | - | 51.6± 0.181 | 43.7± 0.052 | - | - |
| | SD-Net | | | - | 62.0± 0.151 | 52.0± 0.149 | - | - |
| | NoMAD | | | - | 39.5± 0.233 | 29.7± 0.140 | - | - |
| | **CRRL (Ours)** | | | - | **63.5± 0.126** | **57.4± 0.048** | - | - |
| ISO (J) | Stabilizedbci | Direction | 92.5± 0.240 | 57.4± 0.089 | 53.1± 0.205 | 35.0± 0.120 | 36.5± 0.105 | 29.1± 0.015 |
| | SD-Net | | | 65.2± 0.228 | 58.6± 0.159 | 44.2± 0.061 | 43.8± 0.181 | 34.5± 0.032 |
| | NoMAD | | | 65.2± 0.175 | **62.4± 0.023** | 40.8± 0.111 | 37.5± 0.203 | 39.7± 0.119 |
| | **CRRL (Ours)** | | | **66.3± 0.160** | 61.5± 0.204 | **62.2± 0.107** | **48.6± 0.180** | **51.1± 0.008** |
| ISO (S) | Stabilizedbci | Direction | 94.4± 0.072 | 43.3± 0.023 | 47.2± 0.229 | 40.9± 0.172 | 25.0± 0.247 | 14.8± 0.176 |
| | SD-Net | | | 56.5± 0.024 | 53.7± 0.009 | 53.7± 0.060 | 45.6± 0.048 | 35.9± 0.214 |
| | NoMAD | | | 53.3± 0.065 | 51.2± 0.188 | 45.0± 0.127 | 44.1± 0.187 | 31.3± 0.135 |
| | **CRRL (Ours)** | | | **57.2± 0.160** | **56.6± 0.077** | **54.5± 0.237** | **54.1± 0.199** | **45.2± 0.210** |
| ISO (J) | ADAN | EMG | 0.71± 0.060 | 0.66± 0.152 | 0.60± 0.100 | 0.56± 0.110 | 0.54± 0.106 | 0.48± 0.057 |
| | Cycle-GAN | | | 0.68± 0.212 | 0.65± 0.074 | 0.63± 0.114 | 0.59± 0.183 | 0.50± 0.036 |
| | **CRRL (Ours)** | | | **0.70± 0.234** | **0.66± 0.123** | **0.69± 0.223** | **0.65± 0.239** | **0.61± 0.070** |
| Key (G) | ADAN | EMG | 0.46± 0.117 | 0.26± 0.091 | 0.21± 0.019 | 0.18± 0.145 | <0 | - |
| | Cycle-GAN | | | 0.27± 0.112 | 0.25± 0.152 | 0.20± 0.146 | 0.09± 0.234 | - |
| | **CRRL (Ours)** | | | **0.41± 0.193** | **0.37± 0.043** | **0.35± 0.205** | **0.32± 0.101** | - |
| Center-Out (C) | Stabilizedbci | Trajectory | 0.84± 0.186 | 0.59± 0.246 | 0.52± 0.163 | 0.45± 0.189 | - | - |
| | SD-Net | | | 0.73± 0.239 | 0.65± 0.070 | 0.62± 0.176 | - | - |
| | NoMAD | | | 0.55± 0.123 | 0.58± 0.234 | 0.41± 0.223 | - | - |
| | **CRRL (Ours)** | | | **0.75± 0.114** | **0.68± 0.009** | **0.65± 0.106** | - | - |
| Center-Out (M) | Stabilizedbci | Trajectory | 0.68± 0.077 | - | 0.38± 0.199 | 0.34± 0.210 | - | - |
| | SD-Net | | | - | 0.45± 0.060 | 0.39± 0.236 | - | - |
| | NoMAD | | | - | 0.33± 0.036 | 0.26 ± 0.183 | - | - |
| | **CRRL (Ours)** | | | - | **0.51± 0.212** | **0.45± 0.057** | - | - |
| ISO (J) | Stabilizedbci | Trajectory | 0.73± 0.070 | 0.38± 0.239 | 0.32± 0.223 | 0.23± 0.204 | 0.20± 0.117 | 0.17± 0.183 |
| | SD-Net | | | 0.42± 0.228 | 0.35± 0.032 | 0.28± 0.159 | 0.25± 0.157 | 0.20± 0.117 |
| | NoMAD | | | 0.44± 0.163 | 0.45± 0.176 | 0.37± 0.160 | 0.28± 0.127 | 0.31± 0.187 |
| | **CRRL (Ours)** | | | **0.51± 0.246** | **0.48± 0.186** | **0.43± 0.205** | **0.46± 0.043** | **0.47± 0.101** |
| ISO (S) | Stabilizedbci | Trajectory | 0.75± 0.193 | 0.33± 0.146 | 0.35± 0.152 | 0.24± 0.112 | 0.18± 0.237 | <0 |
| | SD-Net | | | 0.40± 0.145 | 0.37± 0.019 | 0.36± 0.091 | 0.32± 0.117 | 0.24± 0.234 |
| | NoMAD | | | 0.34± 0.069 | **0.40± 0.061** | **0.38± 0.136** | 0.37± 0.095 | 0.29± 0.242 |
| | **CRRL (Ours)** | | | **0.43± 0.134** | 0.39± 0.221 | 0.35± 0.187 | **0.37± 0.108** | **0.30± 0.186** |

respectively. Here LR is only set to the LR of the decoder. When comparing one of the parameters, the other parameter is fixed, as shown in the bottom left of Figure 5. Results show that ADAN can fluctuate greatly and even degrade the performance significantly under different parameter Settings. However, our CRRL and Cycle-GAN are more stable. In contrast, our method performs better than Cycle-GAN in all settings.

## B.2 Analysis of the Learned Representations

Visualizing Latent Spaces: We applied t-SNE and PCA to the output of the Reference Alignment (RA) module. These visualizations showed that neural activity from different sessions, after RA transformation, clusters more tightly by behavioral condition or movement type than in the raw neural space. This suggests that the RA module successfully maps inputs from different sessions into a shared, behaviorally meaningful latent space.

Session-Invariance Analysis: We trained a simple linear classifier to predict session identity from the latent representations. The classifier performed close to chance, indicating that the RA outputs carry little information about session-specific features. This supports the idea that the latent space is aligned across sessions and mostly reflects task-relevant structure

## B.3 Analysis of the Sensitivity on Day 0 Data Quality

To evaluate the sensitivity of CRRL under different day 0 qualities, we conducted an experiment in which noise was gradually added to the day 0 signal and observed the changes in performance.

Although CRRL performance is affected by the quality of Day 0 data, using average of multi-day effectively reduces this dependency.

Table 8: The sensitivity of CRRL under different day 0 qualities.

| Noise Level | ISO J [5, 10) | ISO J [10, 20) | ISO J [20, 40) |
|---|---|---|---|
| $\sigma = 0.0$ | **0.51/66.3** | **0.48/61.5** | **0.46/48.6** |
| $\sigma = 0.2 \times std$ | 0.48/62.9 | 0.45/58.0 | 0.42/43.0 |
| $\sigma = 0.4 \times std$ | 0.44/57.5 | 0.41/52.3 | 0.37/36.7 |
| $\sigma = 0.6 \times std$ | 0.39/50.2 | 0.36/45.1 | 0.31/29.6 |

Table 9: Using the average of multiple days as Day 0.

| Noise Level | Number of days | ISO J [5, 10) | ISO J [10, 20) | ISO J [20, 40) |
|---|---|---|---|---|
| $\sigma = 0.2 \times std$ | - | 0.48/62.9 | 0.45/58.0 | 0.42/43.0 |
| $\sigma = 0.4 \times std$ | - | 0.44/57.5 | 0.41/52.3 | 0.37/36.7 |
| $\sigma = 0.6 \times std$ | - | 0.39/50.2 | 0.36/45.1 | 0.31/29.6 |
| $\sigma = 0.2 \times std$ | 2 days | 0.50/64.7 | 0.47/60.2 | 0.44/46.1 |
| $\sigma = 0.4 \times std$ | 2 days | 0.46/60.5 | 0.43/55.4 | 0.39/40.9 |
| $\sigma = 0.6 \times std$ | 2 days | 0.42/54.0 | 0.39/48.8 | 0.37/33.2 |
| $\sigma = 0.2 \times std$ | 3 days | **0.51/65.5** | **0.48/61.0** | **0.45/47.5** |
| $\sigma = 0.4 \times std$ | 3 days | **0.49/61.7** | **0.45/56.8** | **0.44/46.3** |
| $\sigma = 0.6 \times std$ | 3 days | **0.48/59.5** | **0.45/53.7** | **0.41/43.9** |
| $\sigma = 0$ | - | 0.51/66.3 | 0.48/61.5 | 0.46/48.6 |

The table shows that averaging over multiple days—especially 3-day averaging—effectively mitigates the degradation of performance and improves both stability and accuracy.

Table 10: Choosing another day as Day 0.

| Day 0 | Performance on the data after 20 days |
|---|---|
| Jango_20150730 | 0.46/48.6 |
| Jango_20150801 | 0.45/53.0 |
| Jango_20150805 | 0.45/51.1 |
| Jango_20150905 | 0.48/59.2 |

We randomly selected four different days from the Jango dataset as Day 0 and trained and evaluated them separately. We found that models trained on different Day 0 all achieved good cross-day alignment performance which shows that it is easy to find a day 0 with good quality.

## B.4 Additional Ablation Study

### B.4.1 Higher Change Ratios

The results of ablation study on simulation dataset show that when the change ratios increase, the performance of the static decoder will rapidly worsen. However, the performance of CRRL will decline slowly.

### B.4.2 Parameter Sensitivity

In our parameter sensitivity experiments, we scale the parameters of Cycle-GAN and SD-Net and evaluate the regression $R^2$ of EMG and the classification accuracy of directions using the Jango dataset.

Additionally, we reduce the number of parameters in POYO-1. As the number of parameters in POYO decreased, performance gradually declined, which is consistent with the report in the paper of POYO.

### B.4.3 Loss of Rearrangement Learning

Our RA module is designed to learn a permutation matrix that aligns the channels of the re-ordered day k data as closely as possible to those of the day 0 data. Intuitively, using MSE as the loss function like a reasonable choice. However, compared to MSE, the negative Pearson correlation loss focuses more on the shape and trend of the signals rather than their absolute amplitudes, which is particularly important for processing time-series data like neural signals. In our experiments, we

Table 11: Ablation study on simulation dataset with higher change ratios.

| $R^2/acc$ | w/o RA | w/o RC | RA + RC |
|---|---|---|---|
| New/lost neuron (5 %) | 0.83/99.1 | 0.72/92.7 | **0.92/99.5** |
| New/lost neuron (10 %) | 0.81/96.1 | 0.69/89.3 | **0.87/99.3** |
| New/lost neuron (15 %) | 0.65/84.2 | 0.69/89.3 | **0.63/79.5** |
| New/lost neuron (30 %) | 0.43/56.9 | 0.48/51.9 | **0.87/99.3** |
| Shuffled channel (5 %) | 0.15/55.1 | 0.85/92.1 | **0.89/98.0** |
| Shuffled channel (10 %) | 0.05/48.3 | 0.77/87.5 | **0.86/98.2** |
| Shuffled channel (15 %) | <0/25.7 | 0.52/61.3 | **0.67/81.8** |
| Changed function (5 %) | 0.66/80.8 | 0.45/64.7 | **0.73/91.4** |
| Changed function (10 %) | 0.32/56.2 | 0.27/44.2 | **0.55/75.0** |
| Changed function (15 %) | <0/37.0 | <0/28.6 | **0.33/48.4** |

Table 12: The performance of baseline methods on the simulation dataset.

| Type | Method | $R^2/acc$ |
|---|---|---|
| New/lost neuron (10 %) | CRRL | 0.87/99.3 |
| | NoMAD | 0.70/91.4 |
| | SDNet | 0.80/96.3 |
| | Stabilized BCI | 0.76/91.2 |
| Shuffled Channel (10 %) | CRRL | 0.86/98.2 |
| | NoMAD | 0.58/74.0 |
| | SDNet | 0.32/60.9 |
| | Stabilized BCI | 0.18/47.3 |
| Changed Function (5 %) | CRRL | 0.73/91.4 |
| | NoMAD | 0.52/71.5 |
| | SDNet | 0.68/85.5 |
| | Stabilized BCI | 0.51/70.2 |

observed that using negative Pearson correlation loss not only converges more easily but also delivers better performance than MSE.

### B.4.4 Backbone of Reconstruction Module

VQ-VAE discretizes the continuous latent space through vector quantization, forming a discrete codebook. This makes it more advantageous when dealing with data with discrete characteristics (such as spike signals). Specifically, discrete encoding is more robust to input noise and is suitable for handling random disturbances in spike signals. Each vector in the codebook represents a potential pattern, which helps to separate the activation patterns or behavioral states of different channels. Recently, researchers have attempted to use VQ-VAE for encoding on various different signals and have achieved great performance. To further demonstrate the superiority of VQ-VAE in our task, we have conducted a minor experiment to replace VQ-VAE with the original VAE or MLP.

### B.5 The Number of Parameters and Computation Time

The two-module design of CRRL results in extra parameters and computation time compared with regular cross-day methods. However, it still outperforms large-scale pre-training methods.

### B.6 Different Training Processes

End-to-end training involving both discrete permutation learning (RA) and quantized latent variables (RC) was found to be unstable in the experiments. Training RA and RC separately allows us to analyze their behavior better. This also facilitates model reuse and plug-and-play adaptation.

### B.7 The Results of Statistical Tests

We conducted a repeated-measures ANOVA followed by post-hoc Tukey HSD tests across all days on the handwriting dataset, where each method was trained 10 times with different random seeds.

Table 13: The parameter sensitivity experiments.

| Method | Hidden Size | # Params | [5, 10) | [10, 20) | [20, 40) | [40, 65) | [65, 95] |
|---|---|---|---|---|---|---|---|
| Cycle-GAN | Dim=192 | 2.9 M | 0.70±0.14 | 0.64±0.09 | 0.58±0.10 | 0.57±0.07 | 0.49±0.06 |
| | Dim=256 | 4.9 M | 0.68±0.21 | 0.65±0.07 | 0.63±0.11 | 0.59±0.18 | 0.50±0.04 |
| | Dim=384 | 11.1 M | 0.66±0.17 | 0.64±0.12 | 0.60±0.09 | 0.61±0.13 | 0.48±0.06 |
| SD-Net | Dim=192 | 1.1 M | 51.7±0.22 | 49.2±0.14 | 43.7±0.10 | 42.8±0.11 | 33.5±0.06 |
| | Dim=256 | 2.9 M | 65.2±0.23 | 58.6±0.16 | 44.2±0.06 | 43.8±0.18 | 34.5±0.03 |
| | Dim=384 | 7.2 M | 66.1±0.22 | 58.0±0.12 | 43.8±0.10 | 39.2±0.15 | 34.1±0.06 |

Table 14: The parameter sensitivity experiments on POYO-1.

| Method | Hidden Size | # Params | Monkey C |
|---|---|---|---|
| POYO-1 | Dim=128, Layers=24 | 13.0 M | 0.9683±0.01 |
| | Dim=64, Layers=24 | 5.2M | 0.9512±0.0135 |
| | Dim=128, Layers=4 | 3.8 M | 0.9469±0.0109 |
| | Dim=64, Layers=4 | 1.6 M | 0.9403±0.0183 |

We performed post hoc pairwise comparisons using Tukey's HSD test, which corrects p-values for multiple comparisons and controls the family-wise error rate at $\alpha = 0.05$: CRRL > SD-Net (p < .001), CRRL > NoMAD (p < .001), SD-Net > NoMAD (p < .001)

### B.8 The pretraining strategy of CRRL

The choice of a sequential pretraining strategy rather than a more standard joint training approach, such as that used in POYO was based on the following considerations:

Challenges of Joint Training. Our model is designed to learn a permutation matrix that maps neural representations from day k to day 0 within a single monkey. Extending this training regime to multiple monkeys introduces substantial complexity. Simultaneously optimizing permutation matrices for multiple monkeys— without introducing subject-specific modules—would require the model to implicitly resolve both cross-day and cross-subject alignment in a shared parameter space. This significantly increases optimization difficulty and may lead to unstable or suboptimal training.

Structural Consistency. We chose not to modify the core model architecture to accommodate multi-subject joint training. Approaches like POYO often require additional components, such as subject encoders or subject specific normalization, to handle inter-subject variability. Sequential pretraining enabled us to do this without introducing new modules or auxiliary supervision, preserving architectural simplicity and interpretability.

Empirical Support. Sequential transfer learning has been adopted in other machine learning contexts, where models are first pretrained on general data and then fine-tuned to new domains or subjects. Additionally, our experimental results show that sequential pretraining improves performance over w/o pretraining. This demonstrates both the rationality and effectiveness of the approach in our setting.

The goal of CRRL is to achieve long-term stable BCI decoding, so we chose a pretraining method that is easy to implement. Adding more effective pretraining strategies may further improve model performance, which is a promising direction to pursue in future work.

### B.9 Visualization

As shown in Figure 6, we applied t-SNE to the output of the Reference Alignment (RA) module. These visualizations showed that neural activity from different sessions, after RA transformation, clusters more tightly by behavioral condition or movement type than in the raw neural space. This suggests that the RA module successfully maps inputs from different sessions into a shared, behaviorally meaningful latent space.

## C  Related Works

**Cross-Day Neural Decoding.** The key to cross-day neural decoding is eliminating or alleviating the damage of neural signal changes on the decoder which trained on Day 0. The intuitive solution is to align the neural signals from different days. Early studies typically used canonical correlation analysis

Table 15: Ablation Study on Key (G) dataset. NPC means the negative Pearson correlation loss. Replaced means that replace with other channels from a different session.

| $R^2$ | Day 7 | Day 30 | Day 56 |
|---|---|---|---|
| CRRL & MSE | 0.26± 0.050 | 0.18± 0.079 | 0.16 ± 0.088 |
| CRRL & NPC | **0.41± 0.093** | **0.35± 0.007** | **0.32± 0.091** |
| CRRL & VAE | 0.26± 0.105 | 0.21± 0.126 | 0.22± 0.148 |
| CRRL & MLP | 0.36± 0.019 | 0.33± 0.052 | 0.28± 0.091 |
| CRRL & VQ-VAE | **0.41± 0.093** | **0.35± 0.007** | **0.32± 0.091** |
| CRRL wo Replaced | 0.41± 0.093 | **0.35± 0.007** | 0.32± 0.091 |
| CRRL wi Replaced | **0.42± 0.025** | 0.35± 0.091 | **0.32± 0.056** |

Table 16: Number of model parameters and iter speed.

| Method | #Trainable parameters | Iterations per second |
|---|---|---|
| POYO-1 | 13.0 M | 31.91it/s |
| NDT2 | 19.1 M | 11.20it/s |
| SD-Net | 2.9 M | 42.46it/s |
| ADAN | 9.48 MB | 26.72it/s |
| Cycle-GAN | 2.5 M | 21.49it/s |
| CRRL | 6.8 M | 18.25it/s |

(CCA) to align Day 0 and Day k (Altan et al., 2021; Naufel et al., 2019), however, this way can only capture linear changes in neural signals. Recently, there has been a great deal of interest in the concept of a low-dimensional neural manifold (Gallego et al., 2017). For example, (Degenhart et al., 2020) aligns the signals in the neural manifolds obtained by Factor Analysis, NoMAD (Karpowicz et al., 2022) aligns the neural manifolds of Day 0 and Day k in an unsupervised way. While ADAN (Farshchian et al., 2018), Cycle-GAN (Ma et al., 2023), and Dynamic-AE (Fang et al., 2023) use the idea of domain adaptation to align the signals of Day 0 and Day k as source and target domains. Although these methods have achieved good performance, their high sensitivity to perturbations and different parameters limits their ability for clinical application. NDT2 (Ye et al., 2024) and POYO (Azabou et al., 2024) capture the patterns of variation between sessions and sessions by using more data. These methods usually require a large amount of labeled data and are difficult to transfer to other tasks.

**Neural Signal Reconstruction and Imputation.** Reconstructing missing or corrupted neural signals is critical for maintaining decoding accuracy. Autoencoder-based models have been employed to learn robust representations of neural data. (Vincent et al., 2008) introduced denoising autoencoders to reconstruct corrupted inputs by learning latent representations that capture the underlying structure of the data. In the context of neural signals, (Saeed et al., 2021) introduces CHAnnel Reordering Module (CHARM) for EEG data. While our Channel Rearrangement (RA) module also learns channel permutations, the two approaches differ significantly in motivation, assumptions, and implementation. (Liu et al., 2025) proposed BRITS, a method for imputing missing time-series data using bidirectional RNNs with temporal masking. These methods demonstrate the potential of leveraging context information for reconstruction but may not fully exploit inter-channel dependencies in neural signals.

# D Hyperparameters Settings

## D.1 Our CRRL

We undertake several practical considerations to ensure adequate training and inference in our implementation. We adopt the Adam optimizer with a learning rate of 0.001 for model training and utilize a batch size of 128/256 samples. The hyperparameters, such as the weights of different loss $\lambda_{\text{freq}}$ and $\lambda_{\text{vq}}$ are chosen based on validation performance. During masked channel modeling, a masking ratio of 0.05 determines the proportion of channels to mask in each input, and a random channel shuffling rate of 0.1 is applied during the training of the permutation network to enhance robustness.

Table 17: The results of ANOVA.

| Dataset | F | p-value |
|---|---|---|
| Handwriting | 1810.2 | <.001 |

Table 18: The ablation study of different training processes.

| Method | ISO J [5, 10) | ISO J [10, 20) | ISO J [20, 40) |
|---|---|---|---|
| End-to-end training | 0.51/65.8 | 0.41/57.7 | 0.36/40.2 |
| Two-stage training | **0.51/66.3** | **0.48/61.5** | **0.46/48.6** |

The embedding size of encoder $E_\theta$ is 128, and the number of attention heads is 4. The codebook size $K$ is 512, and the commitment cost weight $\beta$ is set to 0.25 to balance expressiveness and computational efficiency. Our model is implemented using PyTorch, enabling flexible model design and training, and training and inference are performed on GPUs to accelerate computation. All experiments were conducted on NVIDIA Tesla A100 GPUs, and the maximum memory usage across all experiments was approximately 22 GB.

## D.2   Stabilizedbci

Stabilizedbci (Degenhart et al., 2020) aims to stabilize brain-computer interfaces (BCIs) by leveraging the concept of a "neural manifold," a low-dimensional representation of population-level neural activity. The proposed approach addresses the challenge of neural recording instabilities, such as electrode shifts or neuron dropout, which can disrupt BCI performance. By aligning the neural manifold across different time points using stable electrodes as reference points, the method maintains a consistent mapping of neural activity to movement intent. This eliminates the need for frequent recalibration, allowing the BCI to provide reliable performance even under severe recording instabilities, and it does so without requiring knowledge of the user's intent.

## D.3   SD-Net

SD-Net (Fang et al., 2023) focuses on extracting both semantic and dynamic features from neural signals, leveraging the idea that low-dimensional dynamics can represent high-dimensional neural activity. By embedding these features into a unified space, the method enables recalibration without requiring labeled data from new sessions. Additionally, the paper introduces a joint distribution alignment strategy, which aligns both marginal and conditional distributions to handle complex domain shifts in neural data. This approach is validated on real and simulated datasets, demonstrating its effectiveness in improving BCI performance across sessions.

## D.4   NoMAD

NoMAD (Nonlinear Manifold Alignment with Dynamics) (Karpowicz et al., 2022) is a novel method to stabilize brain-computer interface (BCI) decoding performance over time. The approach leverages the low-dimensional manifold structure and dynamics of neural activity, using unsupervised learning to align neural data across different time periods. By incorporating recurrent neural network (RNN) models to capture neural dynamics and an alignment network to map neural data to a consistent manifold, NoMAD allows the initial decoder to maintain high accuracy without requiring additional labeled data or frequent recalibration. This method was validated on monkey motor task datasets, demonstrating superior decoding stability and performance compared to existing methods, paving the way for more practical and robust BCI systems

## D.5   Cycle-GAN

Cycle-GAN (Ma et al., 2023) uses Cycle-Consistent Adversarial Networks to align neural activity recorded on different days, enabling a fixed decoder trained on an initial day to maintain accurate predictions without requiring frequent recalibration. Unlike previous methods, Cycle-GAN operates directly on the full-dimensional neural signals, avoiding information loss from dimensionality reduction and offering greater robustness and ease of training. The approach outperforms prior

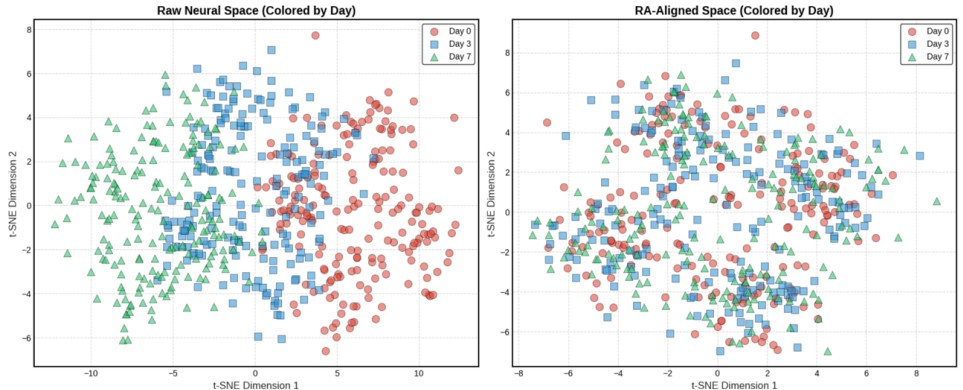

Figure 6: Visualization of latent space.

Table 19: Hyperparameters for rearrangement training.

| Hyperparameters | Values |
|---|---|
| Batch Size | 128 |
| Initial Temperature $\tau_{\text{initial}}$ | 1 |
| Shuffling Rate | 0.1 |
| Final Temperature $\tau_{\text{final}}$ | 0.001 |
| Decay Rate $\gamma$ | 0.01 |
| Entropy Coefficient $\lambda_{\text{entropy}}$ | 0.2 |
| Number of Training Epochs | 300 |
| Learning Rate | 0.001 |

alignment techniques like ADAN and Procrustes-based methods, making it a promising solution for long-term iBCI applications.

### D.6 ADAN

ADAN (Farshchian et al., 2018) combines a deep learning-based neural autoencoder with an EMG predictor to extract a low-dimensional representation of neural signals while simultaneously predicting movement intent. To stabilize the BMI over time, they implement domain adaptation techniques, including Canonical Correlation Analysis (CCA), Kullback-Leibler Divergence Minimization (KLDM), and a new Adversarial Domain Adaptation Network (ADAN). ADAN, which aligns the probability distributions of residuals from neural signal reconstructions, outperforms other methods and requires minimal data to adapt, offering a more robust and consistent interface for users.

### D.7 POYO-1

POYO-1 (Azabou et al., 2024) is a large-scale pre-trained model based on a Transformer architecture designed for neural population decoding, which aims to integrate neural data across experiments, individuals, and laboratories. The core innovation lies in directly converting each neural spike into a Token, preserving the fine temporal structure, and utilizing the PerceiverIO architecture to compress long sequence data to reduce computational cost. The model solves the problem of no correspondence between neurons across sessions by Unit Embedding and Session Embedding, and supports few-shot transfer learning and fast adaptation to new datasets. Trained on a diverse dataset covering 27,373 neural units from seven nonhuman primates, POYO-1 significantly improves decoding performance on complex behavioral tasks such as random object tracking, providing an efficient and scalable unified framework for brain-computer interface and neurodynamics research.

Table 20: Hyperparameters for reconstruction training.

| Hyperparameters | Values |
|---|---|
| Batch Size | 256 |
| Masking Ratio | 0.05 |
| Embedding Size of Encoder | 128 |
| Num of Attention Head | 4 |
| Codebook Size | 1024, 64 |
| Commitment Cost Weight $\beta$ | 0.25 |
| $\lambda_{freq}$ | 0.1 |
| $\lambda_{vq}$ | 0.2 |
| Number of Training Epochs | 500 |
| Learning Rate | 0.001 |

Table 21: Hyperparameters for Stabilizedbci training.

| Hyperparameters | Values |
|---|---|
| Dim of Latent | 10 |
| Num of FA models | 5 |
| Maximum Num of EM iterations | 100000 |
| EM stopping criteria | 0.00001 |
| Minimum private variance threshold | 0.1 |
| Num of rows of loading matrix | 90 |
| Alignment L2 norm threshold | 0.01 |

### D.8   Neural Data Transformer 2 Multi-Context (NDT2 Multi)

NDT2 (Ye et al., 2024) is a Transformer-based neural spiking activity model, which significantly improves the adaptation speed and performance of Brain-Computer Interface (iBCI) decoding tasks through multi-context pre-training across sessions, subjects and tasks, combining spatio-temporal attention and context embedding. NDT2 Multi uses neural data from multiple sessions to jointly train the model to enhance the adaptability of the model to inter-session differences.

## E   Gradient Propagation Analysis for Gumbel-Sinkhorn Networks

### E.1   Problem Formulation

Given a log-probability matrix $\mathbf{L} \in \mathbb{R}^{C \times C}$ which obtained by our channel permutation network, we seek to solve the combinatorial optimization problem:

$$\mathbf{P} = \arg \max_{\mathbf{P} \in \mathcal{P}_C} \langle \mathbf{L}, \mathbf{P} \rangle, \tag{31}$$

where $\mathcal{P}_C$ denotes the set of $C \times C$ permutation matrices. Direct differentiation through the discrete $\arg \max$ operator is infeasible due to its non-smooth nature.

### E.2   Differentiable Reparameterization with Gumbel Noise

Following (Jang et al., 2017b), we inject Gumbel-distributed noise $\epsilon_{i,j} \sim \text{Gumbel}(0, 1)$ into the logits:

$$\tilde{\mathbf{L}}_{i,j} = \mathbf{L}_{i,j} + \gamma \epsilon_{i,j}, \tag{32}$$

where $\gamma > 0$ controls the noise magnitude. This reparameterization preserves the original distributional properties while enabling gradient flow.

Table 22: Hyperparameters for SD-Net training.

| Hyperparameters | Values |
| --- | --- |
| Learning Rate | 0.002 |
| Batch Size | 10 |
| Dim of Latent | 256 |
| Epoch | 60 |
| $\lambda_1, \lambda_2$ | 0.1, 0.1 |
| $\lambda_3, \lambda_4$ | 0.1, 0.1 |
| $\lambda_5, \lambda_6$ | 0.1, 1 |

Table 23: Hyperparameters for NoMAD training.

| Hyperparameters | Values |
| --- | --- |
| Initial learning rate | 10 |
| Batch size | 5 |
| NLL cost weight | 100000 |
| NLL ramping epochs | 0.00001 |
| KL ramping epochs | 0.1 |
| KL weight on initial conditions | 90 |
| KL weight on controller outputs | 0.01 |

## E.3 Continuous Relaxation via Entropy-Regularized Optimal Transport

The Gumbel-perturbed logits are transformed into a doubly stochastic matrix $\mathbf{M}$ through the Sinkhorn-Knopp algorithm (Cuturi, 2013). Let $\tau > 0$ be a temperature parameter:

$$\mathbf{K}^{(0)} = \exp\left(\frac{\tilde{\mathbf{L}}}{\tau}\right). \tag{33}$$

Alternating row and column normalizations are applied for $T$ iterations:

$$\mathbf{R}^{(t)} = \mathrm{diag}\left(\mathbf{K}^{(t-1)}\mathbf{1}_N\right)^{-1}, \tag{34}$$

$$\mathbf{K}^{(t)} = \mathbf{R}^{(t)}\mathbf{K}^{(t-1)}, \tag{35}$$

$$\mathbf{C}^{(t)} = \mathrm{diag}\left(\mathbf{1}_N^\top\mathbf{K}^{(t)}\right)^{-1}, \tag{36}$$

$$\mathbf{K}^{(t)} = \mathbf{K}^{(t)}\mathbf{C}^{(t)}, \tag{37}$$

where $\mathbf{1}_N$ is a column vector of ones. The final doubly stochastic matrix is $\mathbf{M} = \mathbf{K}^{(T)}$.

## E.4 Differentiability of the Sinkhorn Operator

**Proposition E.1** (Implicit Gradient Computation). *The Jacobian $\nabla_{\tilde{\mathbf{L}}}\mathbf{M}$ can be computed via implicit differentiation of the Sinkhorn iterations (Cuturi, 2013; Mena et al., 2018).*

*Proof.* Let $\mathcal{S}(\tilde{\mathbf{L}}/\tau)$ denote the Sinkhorn operator. The gradient chain through $T$ iterations is:

$$\frac{\partial \mathbf{M}}{\partial \tilde{\mathbf{L}}} = \prod_{t=1}^{T} \frac{\partial \mathbf{K}^{(t)}}{\partial \mathbf{K}^{(t-1)}} \cdot \frac{\partial \mathbf{K}^{(0)}}{\partial \tilde{\mathbf{L}}}. \tag{38}$$

Each term $\frac{\partial \mathbf{K}^{(t)}}{\partial \mathbf{K}^{(t-1)}}$ is derived from the row/column scaling operations, which are element-wise differentiable. Full derivations are provided in (Mena et al., 2018). □

Table 24: Hyperparameters for Cycle-GAN training.

| Hyperparameters | Values |
|---|---|
| Batch size | 256 |
| Discriminator (D1) learning rate | 0.01 |
| Discriminator (D2) learning rate | 0.01 |
| Generator (G1) learning rate | 0.001 |
| Generator (G2) learning rate | 0.001 |
| Number of training epochs | 200 |

Table 25: Hyperparameters for ADAN training.

| Hyperparameters | Values |
|---|---|
| Dim of Decoder Latent | 10 |
| Batch Size of Decoder | 64 |
| Epochs of Decoder | 400 |
| Learning Rate of Decoder | 0.0001 |
| Decoder layers | 1 |
| Epochs of ADAN | 200 |
| Batch Size of ADAN | 4 |
| Learning Rate of Discriminator | 0.00001 |
| Learning Rate of Generator | 0.0001 |

## E.5 Convergence to Exact Permutations

**Proposition E.2** (Discrete Limit). *As $\tau \to 0^+$,* $\mathbf{M}$ *converges almost surely to the permutation matrix* $\mathbf{P}$ *obtained by the Hungarian algorithm on* $\tilde{\mathbf{L}}$.

*Proof.* When $\tau \to 0^+$, the entries of $\mathbf{K}^{(0)}$ satisfy:

$$\lim_{\tau \to 0^+} \exp\left(\frac{\tilde{\mathbf{L}}_{i,j}}{\tau}\right) = \begin{cases} 1 & \text{if } (i,j) = \arg\max_{k,l} \tilde{\mathbf{L}}_{k,l} \\ 0 & \text{otherwise} \end{cases}. \tag{39}$$

The Sinkhorn iterations preserve the dominant permutation pattern, recovering the Hungarian solution (Mena et al., 2018). □

Table 26: Hyperparameters for POYO-1 training.

| Hyperparameters | Values |
| --- | --- |
| Num of Token | 512 |
| Hidden Dim | 128 |
| Num of Attention Head | 8 |
| Num of Layers | 24 |
| Len of Context | 1s |
| Epochs | 400 |
| Batch Size | total 1400 |
| Learning Rate of Fine-Tune | 0.0001 |
| Learning Rate of Pretrain | 0.0003 |

Table 27: Hyperparameters for NDT2 Multi training.

| Hyperparameters | Values |
| --- | --- |
| Hidden Dim | 512 |
| Num of Attention Head | 8 |
| Transformer Layers | 6 |
| Learning Rate | 0.0001 - 0.0004 |
| Dropout | 0.1 |
| Early Stopping | 20 |
| Batch Size | 64 |
| Len of Context | 600 ms |

