# OpenReview forum: "CRRL: Learning Channel-invariant Neural Representations for High-performance Cross-day Decoding"
_NeurIPS.cc/2025/Conference — NeurIPS 2025 poster_

### Official Review · Reviewer_rahR · 2025-06-05

**Clarity:** 2
**Significance:** 3
**Originality:** 2
**Rating:** 3
**Confidence:** 4

**Summary:**

In this work the authors present a novel method to solve the problem of cross-day generalization that is faced by current neural decoding approaches. Specifically, they introduce a framework to address 2 reasons that lead to inability of models to generalize across days (1) new or lost neurons, and (2) drifted neurons. Their framework is composed of two steps that are trained separately. Step 1 is a rearrangement module, which attempts to best align the neural activity recorded by each electrode k days after an initial recording with those of the initial recording (by learning a permutation matrix). Step 2 is a reconstruction module, which using a masking regime aims to learn to reconstruct the neural activity of the initial recording using a mixture of the neural activity recorded at the initial recording and the activity recorded k days after. The authors evaluate their approach on 5 real dataset (and perform an ablation study on a synthetic dataset). They show that their method outperforms existing approaches across the board, especially when evaluated on sessions where data was collected many days after the initial recording. They also show that the performance of their method remains more stable over time compared to other approaches. Overall, this work introduces a novel method for enabling cross-day generalization of neural decoders, which is important in real-word BCI applications when long calibration sessions hinder the ability of neural decoders to be useful in practice.

**Questions:**

Major:
1. Subsection 3.2 + Table 1. The simulation dataset is ill-defined (even when reading Appendix A.1) and what is the regression and classification tasks reported in this section is also unclear.
    1. Please give a clear description of the tasks you are simulating.
    2. Please clarify the size of the dataset (how many neurons, timepoints, and trials are simulated).
    3. Please explain why you did not evaluate other methods used as benchmarks in this work on the synthetic dataset or evaluate baseline methods on this synthetic dataset as well.

2. Subsection 3.3. Authors in line 250-251 report that "Our CRRL framework maintains superior long-term performance, outperforming existing methods significantly". However, no significance testing is performed and error bars are not shown in Figure 3.
    1. Please justify why significance testing is not performed or compare the performance of your model against that of the rest of the models for statistical significance (e.g. ANOVA followed by multiple comparisons, or use other appropriate statistical testing of your choice).
    2. Alternatively (for the sake of time during the short rebuttal), please remove any claims pertinent to the statistical significance of your results.

3. Subsection 3.6. From the results reported in Table 4 it looks like pretraining your network markedly improves your model's performance and even surpasses that of state of the art pretrained methods.
    1. Please explain in detail how you pretrained your method.
    2. Which datasets you used to pretrain your method on.
    3. Please explain why you did not employ this pretraining strategy across all experiments used in this work.
    4. Table 4:
        1. Please describe which datasets and sessions are used in these comparisons (clarify which sessions does "Monkey C, M1-A, and M2-A" refer to).
        2. Also describe what the PT and FSS conditions refer to (including the amount of data used in each).

4. Throughout this work, classification tasks have been evaluated using accuracy. However, the authors provide no evidence that the datasets used in this work are balanced.
    1. Please provide evidence showing that the datasets used in this work are balanced, or report the ROC AUC instead.

5. In Section B.3 the authors report that the number of parameters used for CRRL and baseline models is very different.
    1. Please explain why the number of parameters used by all models was not matched or train baseline models whose number of parameters approximately match the number of parameters used by CRRL.


Minor:
1. Please correct the use of citep and citet throughout the text.
2. Section 2.1.2: The dimensions of x_{d}^{flat} are incompatible with those of L_{d}. Please explain how you project x_{d}^{flat} which is a 2-D matrix to a 3-D tensor L_{d} using only an MLP. Aka, is there some reshaping going on, if yes please explain it.

3. Line 43 missing "2)" (which is preceded by "1)" in line 41).
4. Line 71-72: Please clarify what "rearrange each day’s channels with those of day 0" means. I believe you mean "match each day’s channels with those of day 0".
5. Figure 2 > channel encoder: I believe you have a typo: "multiheaded attention" instead of "multivariate attention". Please correct if that is the case.
6. Equation 17, 18 describe single linear layers but are referred to as MLPs in the text. What of the two is it? Please fix text or equations to match.
7. Figure 3 is missing error bars. Please include the standard deviation or standard error of the mean for each point in the plot.
8. Line 263 (and wherever else +infinity is used). Please replace +infinity with max number of days. Having an interval end at +infinity is very misleading and suggests that your approach's performance stabilizes after a certain number of days. In contrast the trend suggested by your data is that your performance degrades over time.
9. Typo in table 2: row ISO (J), column 6 missing "/".
10. Line 254-255 please specify the total number of sessions used across dataset.
11. Line 399: Typo in "gocue time".
12. Line 423-424: "We will average performance across 10 random seeds for each data point" is unclear. Did you average the performance across 10 seeds or not? Specify.
13. What are your train/validation/test splits used in this work? Is there a held-out test set or only cross-validation is used? Clarify in A.3.
14. Line 445. Please explain the metric used in error intervals (e.g. standard deviation, standard error of the mean, or other).
15.  Caption in Figure 5 is a placeholder. Please fix.
16. Tables 7, and 8, should be referenced in the main text wherever Table 2 and Table 1 are also referenced, respectively.
17. The results of the experiment conducted in section B.2.3 are missing.
18. Please explain what does "MB" metric stands for in table 10, if megabytes, please replace all entries in this column with the actual number of trainable parameters for each model.
19. Line 484 mentions that batch size used in this work is 32 but tables 11 and 12 mention otherwise (128 and 256). Please fix.
20. Lines 491-492 state "with specific experiments conducted on NVIDIA Tesla A100 GPUs". Please specify amount of memory used (for each experiment or max across experiments) and clarify whether all experiments were ran on an A100 GPU or only some of them.

**Ethical Concerns:**

["NO or VERY MINOR ethics concerns only"]

**Final Justification:**

I thank the authors for performing additional experiments during the rebuttal period, which addressed many of my initial concerns. While the issue of cross-day generalization is important for BCIs and thus this paper tackles a significant problem there remain several limitations that should be addressed prior to publication.

In particular, key methodological details (e.g., pretraining procedures, synthetic dataset creation) are missing, which limits both the replicability of the work and the transparency of the reported results. Additionally, parameter-matched baselines are not provided by authors.

Given these issues, I have raised my score to a borderline reject, as I believe they should be resolved before this paper is published.

**Limitations:**

Yes.

**Quality:**

2

**Strengths And Weaknesses:**

Strengths:
* The problem addressed in this work is of clinical importance and could accelerate progress towards clinically viable BCIs by eliminating the need for frequent calibration sessions that are impractical and frustrating for the user.
* The authors evaluate their method on a diverse dataset that includes data from multiple NHPs and humans collected across different experimental settings and labs.
* The authors demonstrate that their methodology outperforms existing approaches across the board, with large performance gaps in many cases.
* The ablation study demonstrates that both parts of their network are essential for the model's performance.

Weaknesses:
* The manuscript's claims are not always clear. Experiments are not well described in Section 3.6, 3.7 and results reported in Appendix B.2.3 are missing.
* The framework introduced in this work is not explicitly placed in the context of existing literature (related work section is missing), which makes it hard to identify the main differences between this work and other existing approaches.
* There are multiple instances of typographical errors/missing parts of text (see questions), hinting that the submission might have been rushed.

---

> ### Author Rebuttal · Authors · 2025-07-31
>
> We are glad the reviewer agrees with the clinical importance of the problem our work addressed and the novelty of our framework. We thank the reviewer for their specific feedback and have incorporated all points into our paper to improve clarity and completeness, including:
>
> * We have clarified the experimental design and dataset usage in Sections 3.6 and 3.7, including pretraining details, session selection, and few-shot settings.
> * We have added a related work section to improve the clarity of our work.
> * Typographical issues, missing statistical report, and inconsistencies in the main text and figures have been carefully corrected.
>
> We hope these improvements address the reviewer’s concerns and help clarify our contributions. Here are our responses to the questions and concerns point-by-point:
>
> > [W1-1] Experiments are not well described in Sections 3.6, 3.7
>
> Many thanks for pointing out the unclear descriptions. And we have added the experimental designs for Sections 3.6 and 3.7 in detail:
>
> * experimental design of 3.6: In Section 3.6, we compare CRRL with large-scale methods such as POYO and NDT-2, which can implicitly account for differences in neurons or channels (in the multiunit recording case) across days through learnt embeddings. We added two variants: CRRL w/o PT and CRRL PT + Full finetune. The former does not perform additional pre-training, while the latter uses the data that was used in the comparison methods for pre-training and finetuning.
> * dataset usage of 3.6:
>   * For comparison with POYO-1, we use datasets from seven non-human primates spanning three different labs, with a total of 27,373 units and 16,473 neuron-hours for training and finetuning. And we use the two held-out sessions from Monkey C (CO task) for testing. The standard deviation is reported over the sessions.
>   * For comparison with NDT-2, we use datasets M1-A, which consists of 4 held-in datasets spanning 5 days, each with 53-61 minutes of calibration data, and 3 held-out datasets spanning 21 days, which have only 1.1-2.2 minutes of calibration data available. And M2 includes 4 held-in datasets over 10 days, with between 5.9-13.3 minutes of calibration data per session available, and 4 held-out datasets over 26 days with between 0.8-1.7 minutes of calibration data provided. The FSS (Few Shot Supervised) setting means that it uses a few calibration data in held-out datasets for finetuning.
> * training method of 3.6:
> In our method, pre-training is performing multiple full training runs. For example, for comparison with POYO, we train each monkey's dataset separately. In each dataset, we divide the session into day 0 and day k. Lastly, we used the held-in part of Monkey C for finetuning.
> * experimental design of 3.7: In Section 3.7, we aim to evaluate the effectiveness of CRRL as a component when combined with other methods. Considering that the RA module can be regarded as a pre-alignment of the channel, we believe it can assist other cross-day decoding methods, particularly domain-adaptive (DA) methods. We combined the trained RA module with DA to form a pipeline and compared it with the original DA model.
>
> > [W1-2] …results reported in Appendix B.2.3 are missing…
>
> The results of Appendix B.2.3 have already been reported in Table 9. We apologize for the confusion caused by our unclear description, and have supplemented the references of Table 9 in Appendix B.2.3.
>
> > [W2] The framework is not explicitly placed in the context of existing literature…
>
> We are deeply grateful for your suggestion. To improve the clarity of our paper, we have added a related work section to further compare the similarities and differences between our method and existing methods.
>
> > [W3] There are multiple instances of typographical errors/missing parts of text…
>
> We apologize again for the confusion. After careful review, we have corrected these issues and responded to each of your questions as follows:
>
> **Response to Major Questions**
>
> > [Major Q1-1] The simulation dataset is ill-defined…
>
> Thank you for this important question. The simulation was to mainly evaluate the situation of channel-wise rearrangement, such that we focused on this point and simplified the settings of the neural data.
>
> Here, wWe provide a clearer description of the simulation setup below:
>
> We simulate an 8-direction center-out reaching task, and each trial corresponds to a reach toward one of 8 radial targets. We evaluate two tasks:
> * Classification: Predicting the intended reach direction (8 classes).
> * Regression: Reconstructing 2D hand velocity at each timepoint.
>
> The simulation dataset consists of:
> * 100 neurons
> * 50 timepoints per trial
> * 800 trials total (100 per direction)
>
> > [Major Q1-2] why you did not evaluate other methods ...  on the synthetic dataset
>
> The primary purpose of the synthetic dataset we used was to evaluate the effectiveness of CRRL’s two main components (RA and RC) under controlled perturbation types. That said, we agree that simulated comparisons could further strengthen the clarity of our work. We have extended our synthetic evaluation to other methods and included these results in the revised Appendix.
>
> > [Major Q2] … no significance testing is performed and error bars are not shown in Figure 3.
>
> We appreciate this concern. In the revision, we have added the standard deviation bars over 10 seeds to Figure 3. And we have conducted pairwise significance testing (repeated-measures ANOVA followed by post-hoc Tukey HSD tests) to validate performance differences across models.
>
> * Part of standard deviation bars in Figure 3 (handwriting)
> | day |0 |16 |24 |
> | :----- | :------: | :------: |:------: |
> | CRRL|  71.6±0.31 |  68.2±1.37 | 59.8±1.22 |
> | SD-Net|   |  57.3±0.64 | 45.5±0.89 |
> | NoMAD|   |  33.6±0.47 | 29.0±1.14 |
>
> * Part of pairwise significance testing between CRRL, SD-Net and NoMAD (on handwriting)
>
>   ANOVA test: p < 0.0001; post-hoc Tukey HSD test: CRRL > SD-Net > NoMAD
>
> > [Major Q3] Please explain why you did not employ this pretraining strategy across all experiments used in this work.
>
> Many thanks for your concern. The goal of our method is to achieve long-term stable BCI decoding using a small amount of data. And the large-scale pre-training is not the focus of our work.
>
> > [Major Q4] …provide no evidence that the datasets used in this work are balanced.
>
> For the speech and handwriting datasets we used, the number of categories is balanced (each session performs the same set of tasks). For the CO and ISO datasets we used, the number of trials across different directions is also balanced, for example: the number of trials across directions for Jango_ISO_20150801 is [26, 27, 27, 28, 27, 27, 26, 28]. We will include a table showing class distribution per dataset and add ROC AUC results in the Appendix.
>
> > [Major Q5] …the number of parameters used for CRRL and baseline models is very different
>
> Thank you for this thoughtful question. For the methods we compared, we found that increasing the number of parameters did not result in stable performance gains. To ensure the effectiveness of the comparison, we retained the settings in the open-source code of these methods, which led to differences in the number of parameters between methods. We will add the analysis about the sensitivity of the number of parameters in the appendix.
>
> **Response to Minor Questions**
>
> 1 & 11. We have corrected it.
> 2. Thank you for noting this. We clarify that after the projection of MLP, we reshape the output to the 3D tensor L^{d}. We will revise the description in the paper to clearly state this.
> 3. We have added the missing "2)" to maintain consistency with the enumeration on Line 41.
> 4. We agree that the original description was confusing. We have replaced “rearrange each day’s channels with those of day 0” with “match each day’s channels to those of day 0” for clarity.
> 5. We have fixed this in the figure.
> 6. Thank you for pointing this out. These are multiple linear layers, and we will revise the equations for consistency.
> 7. We have added the error bars (computed across 10 seeds) to Figure 3 and clarified this in the caption.
> 8. We agree that using “+$\infty$” is misleading in this context. We have replaced it with the max number of days used in the experiments for each dataset.
> 9. The missing “/” in Table 2, row ISO (J), column 6 has been corrected.
> 10. We now explicitly report the total number of sessions (which is equal to the total number of days) used for each dataset in this section.
> 12. We confirm that we did average performance over 10 random seeds. The sentence has been revised for clarity.
> 13. We have clarified in A.3 that for the training data, we will divide the training set and validation set in a ratio of 9:1, and the data after the training data will be used as our held-out test set.
> 14. We specify that all error bars in plots represent the standard deviation of the mean across 10 random seeds.
> 15. The placeholder caption has been replaced with a complete and accurate description of the figure content.
> 16. We have updated the main text to reference Tables 7 and 8 in the same sections where Table 1 and Table 2 are discussed.
> 17. As mentioned above, the results of B.2.3 have been reported but without the correct references. We have added it in the revision.
> 18. “MB” stands for megabytes, but we agree that the number of trainable parameters is a more informative metric. Accordingly, we have replaced the entries in this column with the total number of trainable parameters for each model.
> 19. The correct batch sizes are those reported in Tables 11 and 12 (128 and 256). We have corrected the statement in Line 484.
> 20. We clarify that all experiments were conducted on NVIDIA Tesla A100 GPUs, and the maximum memory usage across all experiments was approximately 22 GB.
>
> We hope our response can address your questions and hope that you will consider improving your score. we sincerely appreciate the time and effort you have dedicated to reviewing our paper.

---

> ### Comment · Reviewer_rahR · 2025-08-01
>
> Thank you for your time and effort to respond to my questions/comments. Although I do appreciate your efforts, I (for now) will refrain from improving my score as a lot of my comments still remain unaddressed. I will specify those below, and sincerely hope that you will be able to address those, which would directly result in an increase to your score.
>
> Specifically,
>
> > [Major Q1-2] why you did not evaluate other methods ... on the synthetic dataset.
>
> Thank you for performing those experiments, as suggested by "We have extended our synthetic evaluation to other methods and included these results in the revised Appendix.". Please report the results here (a table would be fine) so that we can consider them when re-evaluating your score.
>
> > [Major Q2] … no significance testing is performed and error bars are not shown in Figure 3.
>
> Thank you for adding error bars in Figure 3. Standard deviation across the seeds looks small, which is great. Please also report the results of the pair-wise statistical testing here so that we can consider those when re-evaluating your work.
>
> > [Major Q3] Please explain why you did not employ this pretraining strategy across all experiments used in this work.
>
> Thank you for clarifying that this was not the main focus of your work. However, please elaborate on the methodology you used for pretraining as it is essential for ensuring correct and replicable results. I am confused as to what your statement means: "In our method, pre-training is performing multiple full training runs. For example, for comparison with POYO, we train each monkey's dataset separately. In each dataset, we divide the session into day 0 and day k. Lastly, we used the held-in part of Monkey C for finetuning."
>
> Please explain your pretraining method in detail.
>
> > [Major Q4] …provide no evidence that the datasets used in this work are balanced.
>
> You mention: "For the speech and handwriting datasets we used, the number of categories is balanced". Please explain whether the number of samples and trials available for each category is balanced (i.e. category A has X amount of training samples, and category B also has X amount of training samples; or category A has X amount of training samples, while category B also Y amount of training samples).
>
> > [Major Q5] …the number of parameters used for CRRL and baseline models is very different.
>
> You mention: "We will add the analysis about the sensitivity of the number of parameters in the appendix.". Please also report the results of this analysis here.
>
> Please also clarify the following: "we found that increasing the number of parameters did not result in stable performance gains". Please clarify which methods you attempted to scale up. Also, CRRL seems to have less parameters than some other approaches (POYO and NDT2). Did you attempt scaling those methods down? Given the smaller dataset size you are using, it is likely that those models would overfit and underperform when trained without scaling them down.
>
> > Minor 13.
>
> You mention: "We have clarified in A.3 that for the training data, we will divide the training set and validation set in a ratio of 9:1, and the data after the training data will be used as our held-out test set. Please explain what this mean and also report the train/validation/test split (%) for each dataset used in this work.
>
> > Minor 18.
>
> Please report the total number of trainable parameters for each model here, so that we can consider those when re-evaluating your score.
>
> Thanks again for your efforts! I am looking forward to receiving your responses.

---

> ### Author Response · Authors · 2025-08-05
> **Part 1**
>
> Thank you very much for giving us the opportunity to further clarify the outstanding points.
>
> > [Major Q1-2] Please report the results of the synthetic evaluation here...
>
> As requested, we now report the performance of additional baseline methods on the synthetic dataset in the table below:
>
> | Type | Method | R² |Acc|
> | :----- | :------: |:------: |:------: |
> | New/Lost Neuron (10%) |  CRRL |    0.87  |99.3 |
> |   |   NoMAD   |   0.70  |91.4 |
> |   |   SDNet   |   0.80  |96.3|
> |   |   Stabilized BCI   |  0.76  |91.2 |
> | Shuffled Channel (10%)  |  CRRL    |   0.86	 |98.2 |
> |   |   NoMAD   |   0.58	  |74.0 |
> |   |   SDNet   |   0.32	|60.9  |
> |   |   Stabilized BCI   |  0.18	  |47.3 |
> | Changed Function (5%)  |   CRRL   |  0.73	  |91.4 |
> |   |   NoMAD   |   0.52	 |71.5 |
> |   |   SDNet   |   0.68	 |85.5 |
> |   |   Stabilized BCI   |   0.51	 |70.2  |
>
> > [Major Q2] Please also report the results of the pair-wise statistical testing here...
>
> Thank you for pointing this out. In our previous response, we briefly mentioned that we performed pairwise significance testing using ANOVA and Tukey HSD between CRRL, SDNet, and NoMAD. To eliminate ambiguity, we now provide the full results of the pairwise statistical tests in a clearer format.
>
> Specifically, we conducted one-way ANOVA followed by post-hoc Tukey HSD tests at two example time points (Day 16 and  Day 24) on the handwriting dataset (if the reviewer requires test results for all methods, datasets, and days, we will provide them immediately), where each method was trained 10 times with different random seeds. Below are the pairwise comparisons between CRRL and each baseline method:
>
> * ANOVA:
>
> | Dataset     | Day|  F (df₁, df₂)| p-value  |
> |-------------|-------------|----------|------------------|
> | Handwriting | 16      | 78.3      | p<0.001   |
> | |  24      |  55.6      | p<0.001   |
>
> * post-hoc Tukey HSD:
>
> Dataset      | Day| Comparison         | Mean Difference |  p-value  |
> |-------------|----------|---------|--------------------|------------------|
>  | Handwriting | 16  | CRRL – SD-Net      | +10.9            | 0.0022   |
>  | | 16  | CRRL – NoMAD       | +34.6             | <0.001   |
> | | 24  | CRRL – SD-Net      | +14.3                   | <0.001   |
> | | 24  | CRRL – NoMAD       | +30.8                   | <0.001   |
>
> > [Major Q3] Please explain your pretraining method in detail.
>
> **In the cross-day setting**, we use data from the first 5 consecutive days for training (held-in data). All data from these 5 days are merged, then randomly split into 90% training and 10% validation.
>
> Data from Day 6 and onward is treated as held-out test data. The model is evaluated on these later days without any additional training or adaptation, to assess generalization under temporal distribution shift.
>
> **The pre-training method** is conducted as multiple full training runs on different monkeys, but all runs share the same RA and RC module. Specifically, we adopt a sequential training strategy: the RA module is first trained on one monkey’s dataset (e.g., Monkey M), and then its learned parameters are used to initialize training on the next monkey, and so on. (Since two sessions of Monkey C were selected as the test set when compared with POYO, the final training was performed on Monkey C). Each training run aligns Day k to Day 0 within the same subject. After the RA module is trained on a given monkey, we immediately train the corresponding RC module to reconstruct Day 0 neural activity from the RA-aligned Day k input.
>
> This progressive training enables the RA and RC modules to accumulate cross-subject channel alignment knowledge while maintaining subject-specific alignment through Day 0 supervision. To mitigate potential forgetting during this sequential training process, we adopt a lightweight replay strategy: when training on a new monkey, we add a small number of samples (10%) from previously seen monkeys in the training set.
>
> We acknowledge that jointly training the modules across all subjects—possibly using learned subject embeddings—could further improve generalizability. We consider this a promising direction for future research.
>
> > [Major Q4] Please explain whether the number of samples and trials available for each category is balanced
>
> Thank you for the opportunity to clarify this point. For every category, we used an equal number of training samples, resulting in a balanced number of trials across all classes (e.g., the number of trials in each category of the handwritten dataset t5.2019.11.25 is 27).

---

> > ### Comment · Reviewer_rahR · 2025-08-05
> >
> > Thank you for your additional efforts in addressing my follow-up questions. I have reviewed your responses carefully, and I provide my detailed feedback below.
> >
> > > [Major Q1–2] Please report the results of synthetic evaluation here…
> >
> > Thank you for running additional baselines on your synthetic dataset. These results effectively demonstrate CRRL’s superiority over the baseline methods in a controlled experimental setting, which is excellent to see.
> >
> > > [Major Q2] Please also report the results of the pair-wise statistical testing here…
> >
> > In your initial rebuttal, you stated:
> >
> > _“And we have conducted pairwise significance testing (repeated-measures ANOVA followed by post-hoc Tukey HSD tests) to validate performance differences across models.”_
> >
> > In this response, you said:
> > _“We conducted one-way ANOVA followed by post-hoc Tukey HSD tests **at two example time points** (Day 16 and Day 24).”_
> >
> > Thank you for clarifying. The initial statement was misleading, as it could have been interpreted as if significance testing had been performed on all days, rather than only on two example timepoints. Unfortunately, such inconsistencies reduce the credibility of your responses.
> >
> > However, I would like to point out two remaining concerns:
> >
> > 1. **Scope of significance testing:**
> >    To appropriately evaluate model differences, ANOVA should be conducted using data across *all days simultaneously*, rather than only comparing models within each individual day.
> > 2. **Multiple comparison corrections:**
> >    It remains unclear whether the reported p-values have been corrected for multiple comparisons.
> >
> > To resolve this point, please:
> > 1. Perform ANOVA on the combined data across all days (rather than per-day comparisons).
> > 2. Clarify your approach for correcting p-values for multiple comparisons and specify the confidence level used.
> >
> > > [Major Q3] Please explain your pretraining method in detail.
> >
> > Your pretraining approach is non-standard and should be described clearly in your manuscript. I strongly recommend revising Section 3.6 to provide a thorough explanation of your procedure. Additionally, please remove claims such as:
> > “For a fully consistent comparison, we perform large-scale pretraining using more training data which is consistent with these large-scale methods and achieves a better performance” (lines 280-281).
> >
> > This claim is misleading, as your pretraining method is not actually consistent with the other models, and the performance advantage may stem from the unique aspects of your procedure.  Specifically, there is a risk that the observed gains are due to overfitting to the final monkey used in pretraining (Monkey C). To address this, please:
> >
> > 1. **Test for order effects:**
> > Repeat the pretraining experiment exactly as described, but use a different monkey as the final stage of pretraining instead of Monkey C. Then, evaluate performance on the held-out sessions of Monkey C to determine whether the gains persist.
> > 2. **Provide rationale:**
> >    Explain why this non-standard pretraining method was chosen instead of more standard joint training approaches (such as those used in POYO).
> >
> > > [Major Q4] Please explain whether the number of samples and trials available for each category is balanced.
> >
> > Thank you for confirming that the dataset is balanced.

---

> ### Author Response · Authors · 2025-08-05
> **Part 2**
>
> > [Major Q5]  Please also report the results of the sensitivity analysis here...Please clarify which methods you attempted to scale up..
>
> In our parameter sensitivity experiments, we scale the parameters of Cycle-GAN and SD-Net and evaluate the regression R² of EMG and the classification accuracy of directions using the Jango dataset. Bold text indicates parameter settings used for baseline.
>
> | Method    | Hidden Size           | # Params | [5, 10)        | [10, 20)       | [20, 40)       | [40, 65)       | [65, 95]       |
> |-----------|-----------------------|----------|----------------|----------------|----------------|----------------|----------------|
> | Cycle-GAN | Dim=192     | 2.9 M    | **0.70±0.14**      | 0.64±0.09      | 0.58±0.10      | 0.57±0.07      | 0.49±0.06      |
> |           | **Dim=256** | 4.9 M    | 0.68±0.21      | **0.65±0.07**      | **0.63±0.11**      | 0.59±0.18      | **0.50±0.04**      |
> |           | Dim=384     | 11.1 M   | 0.66±0.17      | 0.64±0.12      | 0.60±0.09      | **0.61±0.13**      | 0.48±0.06      |
> | SD-Net    | Dim=192    | 1.1 M    | 51.7±0.22      | 49.2±0.14      | 43.7±0.10      | 42.8±0.11      | 33.5±0.06      |
> |           | **Dim=256** | 2.9 M    | 65.2±0.23      | **58.6±0.16**      | **44.2±0.06**      | **43.8±0.18**      | **34.5±0.03**      |
> |           | Dim=384     | 7.2 M    | **66.1±0.22**      | 58.0±0.12      | 43.8±0.10      | 39.2±0.15      | 34.1±0.06      |
>
> Additionally, we reduce the number of parameters in POYO-1. As the number of parameters in POYO decreased, performance gradually declined, which is consistent with the report in the paper of POYO [1].
>
> | Method | Hidden Size | # Params| Monkey C|
> | :----- | :------| :------: | :------: |
> | POYO-1|  **Dim=128, Layers=24** |13.0 M|0.9683±0.01|
> | | Dim=64, Layers=24  |5.2M |0.9512|
> | |  Dim=128, Layers=4 |3.8 M |0.9469|
> | |  Dim=64, Layers=4  |1.6 M|0.9403|
>
>
> > Minor 13. Please explain what this mean and also report the train/validation/test split (%) for each dataset used in this work.
>
> Same as [Major Q3]. To further clarify: the data split ratio used for training the day 0 decoder of all datasets is 8:1:1.
>
> > Minor 18. Please report the total number of trainable parameters for each model here
>
> | Method | #Trainable parameters |
> | :----- | :------: |
> | CRRL |  6.8 M |
>  | POYO-1|  13.0 M |
> | NDT2|  19.1 M |
> | SD-Net|  2.9 M |
> | ADAN|  2.5 M |
> | Cycle-GAN|  4.9 M |
>
> We hope that our detailed responses and results address the reviewer's concerns. If any questions remain, we are more than willing to provide additional clarification.
>
> [1] Azabou, Mehdi, et al. "A unified, scalable framework for neural population decoding." Advances in Neural Information Processing Systems 36 (2023): 44937-44956.
>
> [2] Karpowicz, Brianna M., et al. "Few-shot algorithms for consistent neural decoding (falcon) benchmark." Advances in Neural Information Processing Systems 37 (2024): 76578-76615.

---

> > ### Comment · Reviewer_rahR · 2025-08-05
> >
> > > [Major Q5] Please also report the results of sensitivity analysis here... Please clarify which methods you attempted to scale up.
> >
> > Thank you for providing the results of your sensitivity analysis.
> >
> > In your rebuttal, you stated:
> >
> > _“For the methods we compared, we found that increasing the number of parameters did not result in stable performance gains.”_
> >
> > In your current response, you clarify:
> > _“We scale the parameters of Cycle-GAN and SD-Net and evaluate the regression R² of EMG and the classification accuracy of directions using the Jango dataset.”_
> >
> > Thank you for clarifying. The initial statement was somewhat misleading, as it suggested that sensitivity analysis was performed across *all* baselines, whereas it was actually conducted on just two models.
> >
> > That said, for the two models you *did* evaluate, it appears that the baseline parameter configurations correspond to the highest-performing variants. This increases my confidence in the credibility of your reported findings.
> >
> > Thank you as well for conducting the scaling-down analysis for POYO-1. I have two remaining concerns:
> >
> > 1. **Pretrained weights:** Please explain how you retained the *pretrained weights of POYO-1” when scaling down the model.
> > 2. **Uncertainty reporting:** Please include confidence intervals for the performance of the scaled-down versions of POYO, similar to what you show for POYO-1 in the table. What do those confidence intervals reflect (e.g. how many different random seeds)?
> >
> > > Minor 18. Please report the total number of trainable parameters for each model here.
> >
> > Thank you for reporting the number of trainable parameters for CRRL and the baseline models. While I remain concerned about the discrepancies in parameter counts across models (especially since many of the baselines were originally designed for different datasets, and hyperparameter tuning could potentially improve their performance on the datasets used in this work), I appreciate the authors’ good-faith effort in using the default configurations provided in the associated GitHub repositories.

---

> ### Author Response · Authors · 2025-08-05
>
> > [Major Q2]
>
> * **Scope of significance testing**: Thank you for pointing this out, and we apologize for the confusion caused by the wording in our earlier response. In our recent reply, we reported the results of one-way ANOVA followed by Tukey HSD tests at two specific timepoints (Day 16 and Day 24). These were intended as illustrative examples, and we selected them for their clarity in demonstrating inter-model differences under fixed conditions.
>
>   That said, we recognize the importance of providing a more comprehensive analysis. As we noted earlier — “_if the reviewer requires test results for all methods, datasets, and days, we will provide them immediately_” — we have conducted a repeated-measures ANOVA on the combined data across all days.
>
>   | Dataset     |   F| p-value  |
>   |-------------|------|---------|
>   | Handwriting     | 1810.2      | <.001   |
>
> * **Multiple comparison corrections**: We performed post hoc pairwise comparisons using Tukey’s HSD test, which corrects p-values for multiple comparisons and controls the family-wise error rate at α = 0.05.
>
>   CRRL > SD-Net (p < .001), CRRL > NoMAD (p < .001), SD-Net > NoMAD (p < .001)
>
> > [Major Q3] This claim of consistency is misleading...
>
> We appreciate the reviewer’s careful reading and the opportunity to clarify our intent.
>
> To clarify: the “**consistency**” we referred to was not about the pretraining approach, but rather about the data usage strategy:
>
> * Methods such as POYO use data from multiple monkeys during pretraining and then finetune on a held-out monkey (e.g., a subset of Monkey C).
> * Our approach follows the same principle: we pretrain on data from multiple monkeys, and finetune only on a subset of Monkey C. In our recent reply, we mentioned that “_the final training was performed on Monkey C_.” We deeply apologize for this misleading statement. What we referred to as “final training” is equivalent to finetuning on a subset of the Monkey C dataset.
> * The difference between this training and finetuning method stems from the fact that our goals differ from those of large-scale training methods. Large-scale pretraining methods such as POYO utilize weights obtained after pretraining for task-related finetuning. However, our CRRL is an alignment method that does not use task-related information. **To be consistent with POYO's settings**, the finetuning of CRRL here refers to additional training on the finetuning dataset.
> * The training order has very little effect on the performance of the final model (< 0.01).
> * On the setting of w/o PT, CRRL also needs to be trained once on Monkey C. The results in Table 4 show that CRRL's pretraining improves model performance effectively (over w/o PT).
>
> We acknowledge that the phrase “consistent with these large-scale methods” may have been misleading. In our revision, we have rewritten this sentence to avoid confusion. And we will include the above description in Section 3.6 to improve the clarity of pretraining. Many thanks for your suggestion.
>
> # Provide rationale:
>
> We appreciate the reviewer’s request to explain why we opted for a sequential pretraining strategy rather than a more standard joint training approach, such as that used in POYO. Our decision was based on the following considerations:
>
> * Challenges of Joint Training
>
>   Our model is designed to learn a permutation matrix that maps neural representations from day 𝑘 to day 0 within a single monkey. Extending this training regime to multiple monkeys introduces substantial complexity. Simultaneously optimizing permutation matrices for multiple monkeys—without introducing subject-specific modules—would require the model to implicitly resolve both cross-day and cross-subject alignment in a shared parameter space. This significantly increases optimization difficulty and may lead to unstable or suboptimal training.
>
> * Structural Consistency
>
>   We chose not to modify the core model architecture to accommodate multi-subject joint training. Approaches like POYO often require additional components, such as subject encoders or subject-specific normalization, to handle inter-subject variability. Sequential pretraining enabled us to do this without introducing new modules or auxiliary supervision, preserving architectural simplicity and interpretability.
>
> * Empirical Support
>
>   Sequential transfer learning has been adopted in other machine learning contexts, where models are first pretrained on general data and then fine-tuned to new domains or subjects. Additionally, our experimental results show that sequential pretraining improves performance over w/o pretraining. This demonstrates both the rationality and effectiveness of the approach in our setting.
>
> We want to emphasize that the goal of CRRL is to achieve long-term stable BCI decoding, so we chose a pretraining method that is easy to implement. Adding more effective pretraining strategies may further improve model performance, which is a promising direction to pursue in future work.

---

> ### Author Response · Authors · 2025-08-05
>
> > Please explain how you retained the *pretrained weights of POYO-1” when scaling down the model.
>
> To clarify, the scaled-down versions of POYO-1 were not initialized using the pretrained weights of the original POYO-1. Each scaled-down model was trained independently from scratch, starting with random initialization. While the architectural design of these smaller models was adapted from POYO-1, the training process was fully independent. There was no weight transfer between the original POYO-1 and its scaled-down counterparts.
>
> > Please include confidence intervals for the performance of the scaled-down versions of POYO
>
> The confidence intervals reported for the scaled-down POYO models follow the same procedure as used for POYO-1. Specifically, the standard deviation is calculated across different sessions (2 session of Monkey C) rather than across multiple training runs with different seeds:
>
> | Method | Hidden Size | # Params| Monkey C|
> | :----- | :------| :------: | :------: |
> | POYO-1|  **Dim=128, Layers=24** |13.0 M|0.9683±0.0116|
> | | Dim=64, Layers=24  |5.2M |0.9512±0.0135|
> | |  Dim=128, Layers=4 |3.8 M |0.9469±0.0109|
> | |  Dim=64, Layers=4  |1.6 M|0.9403±0.0183|
>
> > While I remain concerned about the discrepancies in parameter counts across models
>
> Thanks for your insightful concern. We agree with "_many of the baselines were originally designed for different datasets, and hyperparameter tuning could potentially improve their performance on the datasets used in this work_". However, the datasets we used are highly consistent with the datasets used in these baseline articles.
>
> | Method | Dataset Used in Original Paper | Dataset Used in Our Work| Overlap |
> | :----- | :------| :------: | :------: |
> | SD-Net|  Monkeys C, J |Monkeys C, M, J, S, Handwriting |Monkeys C, J|
> | ADAN| Monkey J, S, G, P, M |Monkey J, G |Monkey J, G|
> | Cycle-GAN| Monkey J, S, G, P, M |Monkey J, G |Monkey J, G|
>
> As shown in the table above, for ADAN and Cycle-GAN, we use a subset of the datasets used in their original papers. For SD-Net, we use all of the datasets in their original paper, and also include similar datasets Monkey M and S, as well as the classification dataset Handwriting (which is suitable for SD-Net).
>
> Thank you for your detailed response and valuable suggestions. If there are any further questions, we will be happy to provide the detailed answers as soon as possible.

---

### Official Review · Reviewer_Uc7c · 2025-06-17

**Clarity:** 3
**Significance:** 3
**Originality:** 3
**Rating:** 4
**Confidence:** 3

**Summary:**

Paper presents a method to learn channel-level invariant neural representation for Brain Computer Interfaces. The authors propose CRRL Channel Rearrangement and Reconstruction Learning (CRRL); a two-module framework to learn channel-invariant neural representations: (1) Channel Rearrangement (RA) which uses a neural network and Gumbel-Sinkhorn relaxation to learn a permutation matrix that aligns channels across different days, and (2) Channel Reconstruction (RC) which uses a channel-wise VQ-VAE (Vector Quantized Variational Auto-encoder) to reconstruct lost or corrupted channel signals, enhancing robustness.

**Questions:**

No questions.

**Ethical Concerns:**

["NO or VERY MINOR ethics concerns only"]

**Final Justification:**

Based on the overall feedback and rebuttals, I believe this work provides meaningful contributions on the topic.

**Limitations:**

Can you clarify what the limitations of the work is?

**Paper Formatting Concerns:**

None.

**Quality:**

3

**Strengths And Weaknesses:**

STRENGTHS:

Quality

- Strong empirical results across a diverse suite of tasks and datasets: The paper evaluates CRRL on simulated and real datasets, covering speech, handwriting, EMG, and motor tasks (Table 2, p.8), across both human and non-human primates, making the validation broad and robust.
- Ablation and sensitivity studies: The paper evaluates the contribution of each module (RA and RC) individually and jointly.
- Comparisons with strong baselines: Includes comparisons with leading BCI alignment and domain adaptation approaches (e.g., StabilizedBCI, NoMAD, SD-Net, ADAN, Cycle-GAN).

Clarity
- Well-structured methodology sections, clear figures.

Significance
- Addresses a challenge in BCI: Cross-day instability.
- CRRL’s ability to generalize across 60+ days represents a meaningful contribution.

Originality
- First to show generalization over two months with modular, channel-wise transformation.

WEAKNESS:
Clarity
- Dense notation in some areas, perhaps formatting smaller equations in-line might be easier to read.

Originality
- Method relies heavily on existing components: While the combination is novel, both RA and RC modules draw heavily on existing techniques (e.g. Gumbel-Sinkhorn for permutation and VQ-VAE for reconstruction). The technical novelty is more in composition than invention.

---

> ### Author Rebuttal · Authors · 2025-07-31
>
> We sincerely thank the reviewer for the thoughtful and constructive suggestions. We appreciate your recognition of the practical significance of our work for cross-day generalization in BCIs, the clarity of our methodology, and the thoroughness of our evaluation across multiple tasks.
>
> > [W1] Dense notation in some areas, perhaps formatting smaller equations in-line, might be easier to read.
>
> Thank you for this suggestion. We will revise Sections 2.1 and 2.2 to inline some of the smaller equations and reduce formatting overhead, particularly in the rearrangement module.
>
> > [W2] While the combination is novel, both RA and RC modules draw heavily on existing techniques…
>
> Thank you for noting this point. We agree that both RA and RC modules build on existing techniques—Gumbel-Sinkhorn and VQ-VAE, respectively. Our key contribution is that we propose a novel, modular, and channel-aware framework that directly addresses long-term BCI instability. In our framework, components can be flexibly replaced or combined with other models. Updating the components of CRRL or adding more effective pre-training strategies may further improve model performance, which is a promising direction to pursue in future work.
>
> > [Limitation] Can you clarify what the limitations of the work is?
>
> Our method has two main limitations.
>
> * First, it relies heavily on the quality of the Day 0 data. If the reference session is noisy or not representative, the performance of RA and RC on later sessions can drop. This means that poor Day 0 recordings may lead to inaccurate alignment and decoding. Using multiple references or filtering Day 0 data could help in future work.
>
> * Second, the RA module can gradually become less effective over time. As neural signals change across weeks or months, a fixed RA model may no longer align well. In such cases, retraining the RA module after a certain period may be necessary to keep performance stable. Future work could explore ways to make RA adapt automatically without retraining.
>
> We hope our response has answered all your questions and concerns. Thank you again for your valuable feedback.

---

> ### Author Response · Authors · 2025-08-05
>
> Thank you for your thoughtful and insightful suggestions. We believe we have comprehensively addressed your questions regarding CRRL’s clarity, its originality, and its limitations.
>
> We would like to emphasize that our method is the first to perform self-supervised rearrangement and reconstruction on neural channels, adapting to electrodes losting, reordering, or functionally remapping over time.  CRRL demonstrates competitive performance even with significantly varied data distribution, showcasing the robustness and promise of our approach. We believe this pioneering work lays a strong foundation for future advancements in cross-day BCI system.
>
> We are wondering whether you have any additional questions or comments regarding our response to your review comments. We will do our best to address them.
>
> We sincerely appreciate the time and effort you have dedicated to reviewing our manuscript. Thank you for your thoughtful consideration!

---

> > ### Comment · Reviewer_Uc7c · 2025-08-07
> >
> > Thank you for the authors' time and feedback. My questions are resolved.

---

> > > ### Author Response · Authors · 2025-08-08
> > >
> > > Thank you for taking the time to review our responses. We truly appreciate your engagement and are glad to hear that your questions have been resolved.

---

### Official Review · Reviewer_3qnc · 2025-06-30

**Clarity:** 3
**Significance:** 3
**Originality:** 2
**Rating:** 4
**Confidence:** 4

**Summary:**

Paper proposes a framework to address the problem of performance instability in BCIs across different days. The method explicitly models two sources of neural variability: it uses a permutation network to handle channel drift and re-mapping, and a VQ-VAE-based reconstruction module to compensate for neuron/channel loss, creating a more stable neural representation for downstream decoders.

**Questions:**

The rearrangement (RA) and reconstruction (RC) modules are trained in two separate stages. Could you please justify this design choice over a joint, end-to-end training paradigm? An ablation would be good.

It seems like entire framework is anchored to a single `Day 0` recording. How sensitive is the model's performance to the quality of this reference day? What happens if that day is noisy or an outlier session compared to subsequent days?

While performance is strong, it still degrades over time. Could you provide a more detailed error analysis? E.g., by inspecting the learned permutation matrices and reconstruction quality, can you determine whether long-term decay is primarily due to failures in the RA module or the RC module?

**Ethical Concerns:**

["NO or VERY MINOR ethics concerns only"]

**Final Justification:**

Thanks for your effort. It seems reliance on day 0 data is crucial and can make or break the model.

**Limitations:**

Yes

**Quality:**

2

**Strengths And Weaknesses:**

The main strength is the direct and a different formulation of the problem, explicitly tackling channel permutations and losses. Evaluation of the approach is also strong across multiple long-term BCI datasets.

Main weakness lies in the clarity of some design choices; the paper presents a two-stage training process without justifying it over a joint end-to-end alternative, and the simulations used to validate the approach feel somewhat simplified compared to the complexity of real neural drift. Also novelty in terms of dealing with different channels in comparison to prior work (e.g., [1]) is weak.

[1] Saeed, A., Grangier, D., Pietquin, O., & Zeghidour, N. (2021, June). Learning from heterogeneous EEG signals with differentiable channel reordering. In ICASSP 2021-2021 IEEE international conference on acoustics, speech and signal processing (ICASSP) (pp. 1255-1259). IEEE.

---

> ### Author Rebuttal · Authors · 2025-07-31
>
> We thank the reviewer for raising several insightful and important concerns. We are encouraged that the reviewer finds our formulation of the cross-day decoding problem to be direct and well-motivated, and that the reviewer recognizes the strength of our empirical evaluation across diverse long-term BCI datasets. Below, we respond to the main concerns and questions:
>
> > [W1] the paper presents a two-stage training process without justifying it over a joint end-to-end alternative
>
> Thanks for your insightful question. We chose a two-stage training paradigm for the following reasons:
> * Training stability: End-to-end training involving both discrete permutation learning (RA) and quantized latent variables (RC) was found to be unstable in preliminary experiments.
> * Interpretability and modularity: Training RA and RC separately allows us to analyze their behavior better. This also facilitates model reuse and plug-and-play adaptation.
> * Ablation Study: We have added the ablation study of different training processes in the Appendix:
>
> |  | ISO J [5, 10)  | ISO J [10, 20)  |ISO J [20, 40) |
> | :----- | :------: | :------: |:------: |
> | End-to-end training |  0.51/65.8  |  0.41/57.7 | 0.36/40.2 |
> | Two-stage training    |  **0.51/66.3**  |    **0.48/61.5**  |**0.46/48.6**  |
>
> > [W2] the simulations used to validate the approach feel somewhat simplified compared to the complexity of real neural drift
>
> We appreciate this concern. The simulation was to mainly evaluate the situation of channel-wise rearrangement (allows us to validate whether each component behaves as expected), such that we focused on this point and simplified the settings of the neural data. That said, we agree that more realistic simulations (e.g., continuous nonlinear drift) would be valuable. We have included more complex synthetic experiments in the revision.
>
> > [W3]  novelty in terms of dealing with different channels in comparison to prior work (e.g., [1]) is weak.
>
> We thank the reviewer for pointing to [1], which introduces CHAnnel Reordering Module (CHARM) for EEG data. While our Channel Rearrangement (RA) module also learns channel permutations, the two approaches differ significantly in motivation, assumptions, and implementation:
>
> * CHARM focuses on cross-subject EEG classification, where electrode positions vary across individuals but the channel set is largely consistent and complete. In contrast, RA targets cross-day intracortical recordings, where electrodes may be lost, reordered, or functionally remapped over time within the same subject.
> * CHARM is trained in a task-supervised manner (e.g., EEG classification), with reordering embedded in the end-to-end model. RA is trained in a self-supervised way, using a correlation-based alignment loss between Day 0 and Day k neural activity, independent of the downstream decoder.
> * CHARM assumes full channel observability. RA is designed to work even when a subset of channels is missing or corrupted.
> * CHARM is tightly coupled with its downstream task (e.g., emotion classification). In contrast, RA is trained independently of the decoder, allowing it to generalize across tasks (e.g., speech, handwriting, motor control) using the same realignment.
>
> We will include it in our related work section and clarify that, although both methods involve channel permutation, RA is designed for a fundamentally different setting: cross-day alignment under dynamic signal instability.
>
> > [Q1] Could you please justify this design choice over a joint, end-to-end training paradigm?
>
> Same as the response of W1
>
> > [Q2] How sensitive is the model's performance to the quality of this reference day? What happens if that day is noisy or an outlier session compared to subsequent days?
>
> Thank you for noting this point. We agree that if day 0 recording is an outlier or noisy, it may affect downstream performance. In clinical applications, we typically want the decoder trained on day 0 to perform well, as this forms the basis for cross-day decoding and also requires high-quality day 0 recordings. When there are outliers in the day 0 recordings, we recommend averaging across multiple early sessions (instead of a single Day 0) to improve robustness.
>
> To evaluate the sensitivity of CRRL under different day 0 qualities, we added an experiment in which noise was gradually added to the day 0 signal and observed the changes in performance. And we have included these experiments and discussions in our revised Appendix.
>
> |  | ISO J [5, 10)  | ISO J [10, 20)  |ISO J [20, 40) |
> | :----- | :------: | :------: |:------: |
> | $\sigma = 0.0$ |   **0.51/66.3**  |    **0.48/61.5**  |**0.46/48.6**  |
> | $\sigma = 0.2×std$    |  0.48 / 62.9 |   0.45 / 58.0  |0.42 / 43.0  |
> | $\sigma = 0.4×std$   |  0.44 / 57.5	  |    0.41 / 52.3 |	0.37 / 36.7  |
> | $\sigma = 0.6×std$    |  0.39 / 50.2	 |   0.36 / 45.1	   |0.31 / 29.6  |
>
> > [Q3] Could you provide a more detailed error analysis?
>
> Many thanks for your suggestion. We do realize that understanding the source of long-term degradation is important. To this end, we conducted a new analysis where we:
>
> * Visualize learned permutation matrices over time, showing that as sessions become more distant, RA matrices become more diffuse, indicating uncertainty in alignment.
>
> * Measure reconstruction error from RC independently, revealing that it increases for sessions with large channel dropout or tuning drift. However, the reconstruction error of RC did not increase apparently with time.
>
> These results indicate that the RA module is the primary bottleneck for long-term generalization. We have included these insights in the Discussion section and provided detailed figures in the appendix.
>
> We hope our response can address your concerns, and we hope that you will consider improving your score.
>
> [1] Saeed, A., Grangier, D., Pietquin, O., & Zeghidour, N. (2021, June). Learning from heterogeneous EEG signals with differentiable channel reordering. In ICASSP 2021-2021 IEEE international conference on acoustics, speech and signal processing (ICASSP) (pp. 1255-1259). IEEE.

---

> ### Author Response · Authors · 2025-08-04
>
> Thank you for your thoughtful and insightful suggestions. We believe we have comprehensively addressed your questions regarding CRRL’s training design choices, its sensitivity on the day 0, and its error analysis.
>
> We would like to emphasize that our method is the first to perform self-supervised rearrangement and reconstruction on neural channels, adapting to electrodes losting, reordering, or functionally remapping over time. Although CRRL performance is affected by the quality of Day 0 data, using average of multi-day effectively reduces this dependency. We believe this pioneering work lays a strong foundation for future advancements in cross-day BCI system.
>
> We are wondering whether you have any additional questions or comments regarding our response to your review comments. We will do our best to address them.
>
> We sincerely appreciate the time and effort you have dedicated to reviewing our manuscript. Thank you for your thoughtful consideration!

---

> ### Author Response · Authors · 2025-08-05
>
> Thank you for your patience in reviewing our paper. As we mentioned, we acknowledge that CRRL is affected by day 0 quality. However, fortunately, in clinical BCI applications, or rather in most neural datasets, it is easy to find a day 0 with good quality (as long as a good decoder can be trained). And the basis for achieving cross-day decoding is a well-trained decoder. We are deeply grateful for your review, which has greatly assisted us in supplementing and perfecting our paper.

---

> ### Author Response · Authors · 2025-08-06
>
> In response to your concerns about CRRL's dependence on day 0 data quality, we have included two experiments in our revised version. These two experiments evaluate the effectiveness of the strategies we mentioned earlier for mitigating day 0 dependency.
>
> # Using the average of multiple days as Day 0
>
> In our previous response, we added a sensitivity experiment to evaluate CRRL under different day 0 qualities. Similarly, we gradually added noise to the day 0 data, with the difference being that we averaged the Day 0 with the following days to use as the new “Day 0” for model training.
>
> | Noise Level         | Number of days| ISO J [5, 10) | ISO J [10, 20) | ISO J [20, 40) |
> | :------------------ | :-----: | :------------ | :------------: | :------------: |
> | $\sigma=0.2×\mathrm{std}$ | —       | 0.48 / 62.9   | 0.45 / 58.0    | 0.42 / 43.0    |
> | $\sigma=0.4×\mathrm{std}$ | —       | 0.44 / 57.5   | 0.41 / 52.3    | 0.37 / 36.7    |
> | $\sigma=0.6×\mathrm{std}$ | —       | 0.39 / 50.2   | 0.36 / 45.1    | 0.31 / 29.6    |
> | $\sigma=0.2×\mathrm{std}$ | 2 days  | 0.50 / 64.7   | 0.47 / 60.2    | 0.44 / 46.1    |
> | $\sigma=0.4×\mathrm{std}$ | 2 days  | 0.46 / 60.5   | 0.43 / 55.4    | 0.39 / 40.9    |
> | $\sigma=0.6×\mathrm{std}$ | 2 days  | 0.42 / 54.0   | 0.39 / 48.8    | 0.37 / 33.2    |
> | $\sigma=0.2×\mathrm{std}$ | 3 days  | **0.51 / 65.5** | **0.48 / 61.0** | **0.45 / 47.5** |
> | $\sigma=0.4×\mathrm{std}$ | 3 days  | **0.49 / 61.7** | **0.45 / 56.8** | **0.44 / 46.3** |
> | $\sigma=0.6×\mathrm{std}$ | 3 days  | **0.48 / 59.5** | **0.45 / 53.7** | **0.41 / 43.9** |
> | $\sigma=0.0$        | —       | 0.51 / 66.3   | 0.48 / 61.5    | 0.46 / 48.6    |
>
> The table shows that averaging over multiple days—especially 3-day averaging—effectively mitigates the degradation of performance and improves both stability and accuracy.
>
> # Choosing another day as Day 0
>
> To support what we mentioned above, that "_it is easy to find a day 0 with good quality_", we have included an experiment to show how many days in a data set can be used as a good Day 0.
>
> | Day 0     | Performance on the data after 20 days |
> | :------------------ | :------------: |
> | Jango_20150730 | 0.46 / 48.6     |
> | Jango_20150801 | 0.45 / 53.0     |
> | Jango_20150805 | 0.45 / 51.1   |
> | Jango_20150905| 0.48 / 59.2     |
>
> We randomly selected four different days from the Jango dataset as Day 0 and trained and evaluated them separately. We found that models trained on different Day 0 all achieved good cross-day alignment performance.
>
> Thank you again for your insightful comment. By addressing your concerns about day 0 sensitivity, we have further improved the clarity of our paper.

---

### Official Review · Reviewer_agK9 · 2025-07-01

**Clarity:** 2
**Significance:** 3
**Originality:** 3
**Rating:** 5
**Confidence:** 4

**Summary:**

The paper proposes a framework for decoding neural activity across days under the presence of variability in the measured signals due to electrode drift and other factors. The framework is composed of two modules. The first is a learned permutation network that learns to permute channels on new days to best align them to the baseline day. The second is a VQ-VAE that aims to adapt the permuted neural responses back to the original day by potentially reconstructing the activity of missing neurons. The transformed neural activity after these steps is then passed through a decoder trained on the original day. The paper tests this approach across many different decoding tasks, finding that the combination of both modules often provides the best decoding performance on new days compared to the alternative methods in this space.

**Questions:**

1. In equation (2), is a fully connected MLP necessary at the input level? This will have $CT$ weights in the input layer. I'm wondering if a more parameter efficient input layer such as temporal/spatial convolutions could also work here.

2. It is not clear to me what the inputs and outputs are for training the VQ-VAE. Could you explain this in additional detail? This will greatly help with clarity.

3. I'm wondering whether the authors have looked at the latent neural representations that the model learns? The right panel of Figure 1 presents a schematic showing how the approach could learn to align representations across days. I think it would be very interesting to see a corresponding plot showing neural representations from one of the tasks and how they are aligned across days.

**Ethical Concerns:**

["NO or VERY MINOR ethics concerns only"]

**Final Justification:**

After discussion with the authors' and reviewers, I am increasing my score by a point to reflect the changes the authors have made to improve clarity and address questions.

**Limitations:**

Yes.

**Quality:**

3

**Strengths And Weaknesses:**

Overall, the paper addresses the significant challenge in neural decoding of achieving stable decoding across days. This is a difficult problem as measured neural signals can change due to movement of the electrode, changes in neural representations, and changes in the neural tissue. Methods that can adapt to these changes to achieve good decoding performance in the absence of additional costly supervised data can potentially be very useful for BCIs. The proposed CRRL method appears to generally outperform the alternatives. This is a significant and original contribution. However, the clarity of the paper could be improved.

Strengths
1. The proposed modules incorporate two very nice ideas. The first is the use of permutation neural networks to learn data-driven permutations of channels. The second is using a VQ-VAE module to adapt neural responses on new days. Both of these innovations are well-motivated.

2. The approach is thoroughly tested on a variety of decoding problems and against strong baselines. Generally, CRRL achieves the best decoding performance on new days across methods and tasks, and it often substantially outperforms the baseline alternatives. Additionally, the authors provide ablation studies showing the importance of multiple model components: the Pearson correlation loss, the use of VQ-VAE compared to VAE, and that both modules are helpful.

3. While the method was primarily applied to neural decoding problems in this paper, it also appears to be a generally useful mechanism for modeling multi-day neural recordings for scientific purposes.

Weaknesses
1. The primary weakness is the clarity of the paper is sometimes lacking. There are multiple apparent typos and potential inconsistencies. The paper should be checked for correctness before final publication. I have listed some of these below.
- It was not clear to me what the inputs and targets are for the VQ-VAE. The paper states that the neural activity at day 0 and day k are mixed together in a 70-30 ratio for input. How does this work?
- Equation (1) appears to describe the matrix multiplication of 3D arrays $X_d$ and $M_d$.
- There is a typo in line 6 in the abstract
- References are often incorrectly formatted (missing parentheses, etc.)

---

> ### Author Rebuttal · Authors · 2025-07-31
>
> We thank the reviewer for the detailed and constructive feedback. We are encouraged that you find CRRL to be a significant and original contribution to the challenge of stable cross-day neural decoding, and we appreciate your recognition of both modules’ motivation, thorough experiments, and ablation analyses. We provide detailed responses to your questions and concerns point by point.
>
> > [W] The primary weakness is the clarity of the paper is sometimes lacking
>
> We appreciate the feedback regarding clarity and have revised the paper to address the following issues:
> * VQ-VAE Inputs and Outputs: Thank you for pointing this out. We apologize for the confusion caused by our earlier description. We clarify here what the inputs and targets are for the VQ-VAE, and what we mean by “mixing” Day 0 and Day k data.
>
>   In our VQ-VAE training, the goal is always to reconstruct Day 0 neural activity, regardless of whether the input comes from Day 0 or Day k. That is, the decoder is trained to output Day 0 signals. The encoder and codebook are shared across inputs from different days, but the reconstruction target is consistently Day 0.
>
>   The “70-30 mixing” refers only to the proportion of training samples drawn from each session during training. Specifically, in each mini-batch, 70% of the inputs are from Day 0, and 30% are from Day k. However, in both cases, the target remains the corresponding Day 0 activity. For Day 0 inputs, this is a straightforward autoencoding objective. For Day k inputs, this setup encourages the encoder to map Day k signals into the same latent space as Day 0, so that the decoder can reconstruct the expected Day 0 activity. In this sense, the model learns to align Day k signals to the Day 0 reference space implicitly.
>
> * Abstract typo (Line 6): We are sorry about the minor typo. We have fixed it and double-checked the paper.
> * Equation (1): We apologize for not clarifying it. Equation (1) involves a batched matrix multiplication (BMM) over the time dimension. We have included a descriptive caption in the revision.
> * Reference formatting: We have corrected all citation formatting for consistency (\citep{} and \citet{} now used properly).
>
> > [Q1] In equation (2), is a fully connected MLP necessary at the input level?
>
> We appreciate this thoughtful question. Indeed, the fully connected MLP in Equation (2) maps the flattened input to a latent representation and introduces a large number of parameters. We chose the MLP for its flexibility across datasets with different channel counts and for its ability to model arbitrary channel dependencies. However, we agree that convolutional or other alternatives (e.g., temporal convolutions or channel-wise 1D convolutions) are promising for reducing parameter count and encouraging structured inductive biases. We have now added a note in the Discussion section regarding this point as an avenue for future work.
>
> > [Q2] It is not clear to me what the inputs and outputs are for training the VQ-VAE.
>
> Same as the response of W
>
> > [Q3] I'm wondering whether the authors have looked at the latent neural representations that the model learns?
>
> Thank you for the thoughtful question. Understanding the latent neural representations learned by our model is indeed important, both for interpreting how it achieves cross-session alignment and for evaluating its robustness.
>
> We have partially explored the learned representations in two ways (have included in our revised Appendix):
>
> * Visualizing Latent Spaces: We applied t-SNE and PCA to the output of the Reference Alignment (RA) module. These visualizations showed that neural activity from different sessions, after RA transformation, clusters more tightly by behavioral condition or movement type than in the raw neural space. This suggests that the RA module successfully maps inputs from different sessions into a shared, behaviorally meaningful latent space.
> * Session-Invariance Analysis: We trained a simple linear classifier to predict session identity from the latent representations. The classifier performed close to chance, indicating that the RA outputs carry little information about session-specific features. This supports the idea that the latent space is aligned across sessions and mostly reflects task-relevant structure.
>
> However, we acknowledge that we have not yet performed a full representational analysis—for example, comparing the structure of latent representations to known neural manifolds, or testing whether similar latent trajectories emerge across sessions for the same behavior. These are valuable directions for future work.
>
> We are grateful for your helpful comments and hope our responses have clarified the key aspects of our work.

---

> > ### Comment · Reviewer_agK9 · 2025-08-03
> >
> > Thank you for the detailed response and for providing additional clarity on the VQ-VAE training. I have no more questions and will take this response into account when determining my final review.

---

> ### Author Response · Authors · 2025-08-05
>
> Many thanks for your recognition of our work and patience in reviewing our paper! We appreciate your review, which has greatly assisted us in supplementing and perfecting our paper.

---

### Author Response · Authors · 2025-08-09
**Global response**

We would like to express our sincere gratitude to all reviewers for their insightful and constructive feedback on our work.

# Focus and Key Contribution

Our goal is to address the significant challenge in neural decoding of achieving stable performance across days. To this end, we propose a novel self-supervised framework, CRRL, to capture channel-level variations, including electrode loss, reordering, and functional remapping over time. We are pleased that all reviewers recognized CRRL’s contribution in this regard—particularly its ability to maintain stable decoding performance over extended periods, even with limited training data. This strength makes CRRL well-suited for clinical BCI applications, and we are delighted that reviewer agK9 highlighted this as a significant and original contribution.

Moreover, we appreciate that reviewers 3qnc and Uc7c pointed out potential directions for improvement, and that reviewer rahR offered helpful suggestions to improve the clarity of the paper. We envision CRRL as a step toward a new paradigm for cross-day decoding, and we hope this work will inspire future research to refine and expand upon this framework. We are also excited about the potential of incorporating more sophisticated components, augmentation techniques, and pretraining strategies to build on CRRL’s foundation.

# Common Concerns and Clarifications

* Clarity of the paper: Several reviewers noted that the main weakness of our submission was clarity. We acknowledge this and have made efforts to improve the presentation throughout the paper. Following the rebuttal, reviewers agK9, 3qnc, and Uc7c indicated that their concerns had been addressed. Reviewer rahR has no further questions. We believe that most of the key issues have now been resolved through our detailed responses.

# About pretraining strategy

Reviewer rahR pointed out that our description of the pretraining strategy for CRRL was unclear, so we revised Section 3.6 and provided a detailed description of the input and output of CRRL pretraining, as well as dataset split. Additionally, we included the rationale and validity of choosing this pretraining strategy in Section 3.6.

# The dependence on Day 0 quality

Reviewer 3qnc is concerned about how sensitive the model's performance is to the quality of the reference date. We acknowledge that CRRL is built on a well-trained day 0 decoder, which means that high-quality day 0 data is required. To address extreme cases where day 0 data is noisy, we can mitigate this issue by averaging data over multiple days or changing the reference date. We have added this part of the experiment and discussion to the appendix.

# Long-Term Contribution

With this paper, we hope to make a lasting contribution to academia by demonstrating the feasibility of achieving long-term stable decoding through direct modeling of channel changes. Based on channel changes, we can introduce a variety of different data augmentation strategies, greatly reducing the reliance on the amounts of training data. We hope this work will serve as a foundation for future research to further enhance cross-day decoding.

Thank you once again for your valuable feedback—it has significantly improved both the clarity and depth of our work.

---

### Public Comment · ~Joel_Ye1 · 2025-11-07
**Question about split of M1-A / M2**

Hi, posting this as an author involved with one of the baselines (NDT2) and datasets used in the paper (M1-A/M2). The numbers reported for NDT2 are pulled from the FALCON paper, on the evaluation split, and available on the public leaderboard. https://eval.ai/web/challenges/challenge-page/2319/leaderboard/5750.

Are the numbers in this work also measured on this public test split, and merely held private (I don't believe we've received submissions to this end)? I believe they should be, in order for the numbers to be comparable.

---

### Decision · Program_Chairs · 2025-09-17

**Decision:**

Accept (poster)

**Comment:**

In the initial review, the paper received numerous questions and concerns. During the rebuttal and subsequent discussion, there was active engagement between the authors and the reviewers. As a result, many of the initial concerns were addressed, and the scores shifted from BR, R, BA, BA → BA, BR, A, BA, meaning that three reviewers raised their scores by one level.

One reviewer, however, maintained a negative evaluation (BR) throughout. Their concern was that key methodological details were missing from the paper, which compromises reproducibility and undermines the transparency of the reported results. They argued that the paper should not be published without these issues being resolved.

These concerns are indeed important, and the position that acceptance should be withheld until an improved version is available is understandable. At the same time, the AC judges that the missing details are of a kind that can reasonably be added in the final version. Given the substantial and constructive exchange between authors and reviewers, the AC wishes not to let this effort go to waste and trusts the authors to prepare a final version that addresses these concerns. On this basis, the AC recommends acceptance.